# SARS-CoV-2 virulence factor ORF3a blocks lysosome function by modulating TBC1D5-dependent Rab7 GTPase cycle

Kshitiz Walia[1,2], Abhishek Sharma[1], Sankalita Paul[3], Priya Chouhan[1,2], Gaurav Kumar [1], Rajesh Ringe[1], Mahak Sharma [3] & Amit Tuli [1,2] ✉

SARS-CoV-2, the causative agent of COVID-19, uses the host endolysosomal system for entry, replication, and egress. Previous studies have shown that the SARS-CoV-2 virulence factor ORF3a interacts with the lysosomal tethering factor HOPS complex and blocks HOPS-mediated late endosome and autophagosome fusion with lysosomes. Here, we report that SARS-CoV-2 infection leads to hyperactivation of the late endosomal and lysosomal small GTP-binding protein Rab7, which is dependent on ORF3a expression. We also observed Rab7 hyperactivation in naturally occurring ORF3a variants encoded by distinct SARS-CoV-2 variants. We found that ORF3a, in complex with Vps39, sequesters the Rab7 GAP TBC1D5 and displaces Rab7 from this complex. Thus, ORF3a disrupts the GTP hydrolysis cycle of Rab7, which is beneficial for viral production, whereas the Rab7 GDP-locked mutant strongly reduces viral replication. Hyperactivation of Rab7 in ORF3a-expressing cells impaired CI-M6PR retrieval from late endosomes to the trans-Golgi network, disrupting the biosynthetic transport of newly synthesized hydrolases to lysosomes. Furthermore, the tethering of the Rab7- and Arl8b-positive compartments was strikingly reduced upon ORF3a expression. As SARS-CoV-2 egress requires Arl8b, these findings suggest that ORF3a-mediated hyperactivation of Rab7 serves a multitude of functions, including blocking endolysosome formation, interrupting the transport of lysosomal hydrolases, and promoting viral egress.

Severe acute respiratory syndrome coronavirus 2 (SARS-CoV-2), the agent responsible for coronavirus disease 2019 (COVID-19), is a positive sense single-stranded RNA virus belonging to the beta-coronavirus (β-CoV) genus. Positive-sense RNA viruses are known to modify endomembrane compartments to produce replication compartments (RCs). These RCs contain viral RNA and proteins, in addition to host proteins and lipids, and provide a barrier between viral replication and the host cytosol, which contains the viral RNA degradation machinery and innate immune sensors[1,2]. The viral particles formed from structural proteins assemble in the ER-Golgi intermediate compartment. Ultimately, virions transit in vesicles and undergo exocytosis via the biosynthetic secretory pathway[3]. Interestingly, another mechanism for extracellular release, in which virions reside in deacidified lysosomes that fuse with the plasma membrane has recently been reported for β-CoVs, including SARS-CoV-2[4].

The genomic size of SARS-CoV-2 ranges from 29.8 kb to 29.9 kb, encompassing eleven genes with open reading frames (ORFs)[5,6]. The 5′ genomic region constitutes more than two-thirds of the genome and

[1]Division of Cell Biology and Immunology, CSIR-Institute of Microbial Technology (IMTECH), Chandigarh, India. [2]Academy of Scientific and Innovative Research (AcSIR), Ghaziabad, Uttar Pradesh, India. [3]Department of Biological Sciences, Indian Institute of Science Education and Research (IISER), Mohali, Punjab, India. ✉e-mail: atuli@imtech.res.in

encodes the ORF1ab polyproteins. In contrast, the 3' region included genes encoding structural proteins, namely spike (S), envelope (E), membrane (M), and nucleocapsid (N) proteins. Additionally, SARS-CoV-2 contains six accessory proteins encoded by ORF3a, ORF6, ORF7a, ORF7b, ORF8, and ORF10. ORF3a is the largest protein encoded among the SARS-CoV-2 accessory proteins. SARS-CoV-2 ORF3a shares 90% and 72% sequence similarity and identity, respectively, with ORF3a of SARS-CoV-1[7]. Structurally, the 275-amino acid-long ORF3a protein possesses three transmembrane (TM) helices and a cytosolic domain with multiple β-strands in each protomer chain[8]. ORF3a can undergo dimerization, tetramerization, and high-order oligomerization of 31 kiloDaltons (kDa) via intramolecular disulfide bridging[9,10].

Because of its potential involvement in various stages of the viral life cycle, the ORF3a protein of SARS-CoV-2 appears to be an essential virulence factor for driving overall viral pathogenesis. A recent study showed that the deletion of SARS-CoV-2 ORF3a resulted in decreased mortality, a decreased lung viral titer, and reduced tissue damage in K18 hACE2 transgenic mice[11]. Another study also showed that the deletion of ORF3a and E from SARS-CoV-2 results in nonviable virion particles, suggesting a role for ORF3a in the production of infectious virus particles[12]. ORF3a from both SARS-CoV-1 and SARS-CoV-2 was annotated as viroporin, i.e., a viral membrane protein that forms non-selective channels in the plasma membrane or organelle membranes[8,9,13,14]. However, a recent study solved the cryo-EM structures of both SARS-CoV-1 and SARS-CoV-2 ORF3a and revealed that the structure contains a narrow constriction in the transmembrane region and a positively charged aqueous vestibule, which does not favor cation permeation. Thus, the study concluded that the ORF3a proteins of SARS-CoV-1 and SARS-CoV-2 are not ion channels[15].

Previous studies have shown that ORF3a localizes to late endosomes and lysosomes[16,17]. ORF3a contains a YXXΦ based sorting motif (160–163 amino acids) and a double-glycine (diG) region (187–188 amino acids) that mediate its export from the Golgi and determine ORF3a localization to late endosomes and lysosomes[18]. Notably, SARS-CoV-1 ORF3a also contains a YXXΦ motif (160–163 amino acids) that mediates its transport from the Golgi apparatus to the plasma membrane[19]. ORF3a interacts with and sequesters Vps39, a subunit of the heterohexameric lysosomal tethering complex known as the HOPS complex. ORF3a blocks the interaction between the HOPS complex and the autophagosomal SNARE protein syntaxin17 (STX17) and inhibits the assembly of the STX17-SNAP29-VAMP8 SNARE complex, which mediates the fusion of autophagosomes and lysosomes[16]. ORF3a also disrupts the interaction of Vps39 with the Rab7 GTPase, resulting in the failure to form the Rab7-HOPS-SNARE complex, thus blocking the fusion machinery of autophagosomes with LEs and lysosomes[20]. A recent study showed that ORF3a promotes lysosome exocytosis by mediating lysosomal targeting of the BORC complex and the small GTPase Arl8b, which mediate lysosomal motility toward the periphery of the cells[17]. Moreover, ORF3a-expressing cells exhibit elevated cytosolic $Ca^{2+}$ concentrations, which triggers TRPML3-mediated fusion between lysosomes and the plasma membrane[17]. As previously described, β-CoVs, including SARS-CoV-2, bypass the traditional biosynthetic secretory pathway and egress from cells via lysosomal exocytosis[4]. ORF3a-mediated lysosomal exocytosis likely also promotes viral egress. Indeed, ORF3a expression is sufficient to promote extracellular release of the coronavirus MHV-A59, which also employs lysosomal exocytosis for egress[17].

In a previous study from our laboratory, we found fusion machinery at the interface of late endosomes, autophagosomes, and lysosomes. This machinery consists of Rab7-PLEKHM1 on late endosomes, Arl8b, and the HOPS complex on lysosomes[21]. Here we report that by binding to Vps39, ORF3a disrupts the interaction between the HOPS complex and PLEKHM1 and enhances the interaction between PLEKHM1 and Rab7 on late endosomes. We further revealed that the localization of ORF3a to lysosomes and its interaction with Vps39 led

to a dramatic increase in the level of activated Rab7 in host cells. We found that Vps39 interacts with the Rab7 GAP TBC1D5, and that the presence of ORF3a promotes the formation of the Vps39-TBC1D5 complex and the concurrent displacement of Rab7 from this complex. Thus, ORF3a disrupts the GTPase cycle of Rab7 by reducing the interaction between Rab7 and its GAP. We confirmed our findings in SARS-CoV-2-infected cells and found that ORF3a expression was required for Rab7 hyperactivation in infected cells. We observed Rab7 hyperactivation in naturally occurring ORF3a variants encoded by distinct SARS-CoV-2 strains, indicating a link between ORF3a-mediated Rab7 activation and viral growth and survival. Indeed, we found that a constitutively active mutant of Rab7 promoted SARS-CoV-2 replication, while the viral load was strongly reduced in cells expressing a constitutively inactive form of Rab7. The expression of constitutively active Rab7 rescued the SARS-CoV-2 replication defect in ORF3a-depleted cells, indicating that a key role of ORF3a is to promote Rab7 activation, which in turn is required for viral replication.

ORF3a blocked the retrieval of CI-M6PR from Rab7-positive endosomes to the TGN and, consequently, impaired the sorting of newly synthesized lysosomal hydrolases from the TGN. ORF3a expression and SARS-CoV-2 infection also impaired the fusion of Rab7-positive compartments with Arl8b and LAMP1-positive compartments that mediate virus egress from host cells. Taken together, our results suggest that SARS-CoV-2, through its accessory protein ORF3a, blocks Rab7 GTP-GDP cycling, which impairs lysosome fusion with other compartments and rather promotes the egress of the virus via lysosomal exocytosis.

## Results

### ORF3a localizes to lysosomes, and its expression leads to the formation of enlarged perinuclear lysosomes

To investigate the impact of SARS-CoV-2 ORF3a expression on lysosomal function, we first generated a stable, tetracycline-inducible, streptavidin-tagged ORF3a-expressing HeLa cell line (HeLa$^{ORF3a}$) or transiently transfected ORF3a with a C-terminal streptavidin, GFP, or HA tag. Before investigating the phenotype associated with ORF3a expression, we confirmed whether ORF3a levels in the HeLa$^{ORF3a}$ stable cell line were comparable to those observed upon SARS-CoV-2 infection. To this end, we procured an antibody against ORF3a and confirmed its specificity by siRNA-mediated ORF3a knockdown in SARS-CoV-2-infected Vero E6 cells (Supplementary Fig. S1A). Using this antibody, we found that ORF3a levels in the HeLa$^{ORF3a}$ stable cell line after 24 h of doxycycline (Dox) treatment (1 μg/mL) were similar to those observed in SARS-CoV-2-infected Vero E6 cell lysates, confirming that the phenotypes observed in the HeLa$^{ORF3a}$ stable cell line will be physiologically relevant (Supplementary Fig. S1B). Thus, we treated cells for 24 h with Dox (1 μg/mL) for all the experiments using the stable HeLa$^{ORF3a}$ cell line.

We first examined the subcellular localization of ectopically expressed ORF3a and found that it strongly colocalized with Lysotracker-positive puncta, as well as with LAMP1 and cathepsin D, confirming its previously reported localization to lysosomes (Fig. 1A, Supplementary Figs. S1C, S1D and quantification of the Pearson Correlation Coefficient (PCC) shown in Fig. 1D and Supplementary Fig. S1F). We also found a modest overlap of ORF3a with early endosomes that were enlarged in ORF3a-expressing cells but not in the Golgi or recycling endosomes (Fig. 1B, C, Supplementary Fig. S1E, and quantification of the PCC is shown in Fig. 1D and Supplementary Fig. S1F). In A549 cells, we observed a similar localization of ORF3a to LAMP1-positive endosomes (Supplementary Fig. S1G). Consistent with the localization of epitope-tagged ORF3a, partial colocalization of ORF3a and LAMP1 was also observed in SARS-CoV-2-infected Vero E6 cells immunostained with an anti-ORF3a antibody (PCC ~ 0.34 ± 0.05), as shown in Supplementary Fig. S1H. Using a standard protocol for lysosome enrichment based on Optiprep-based density gradient

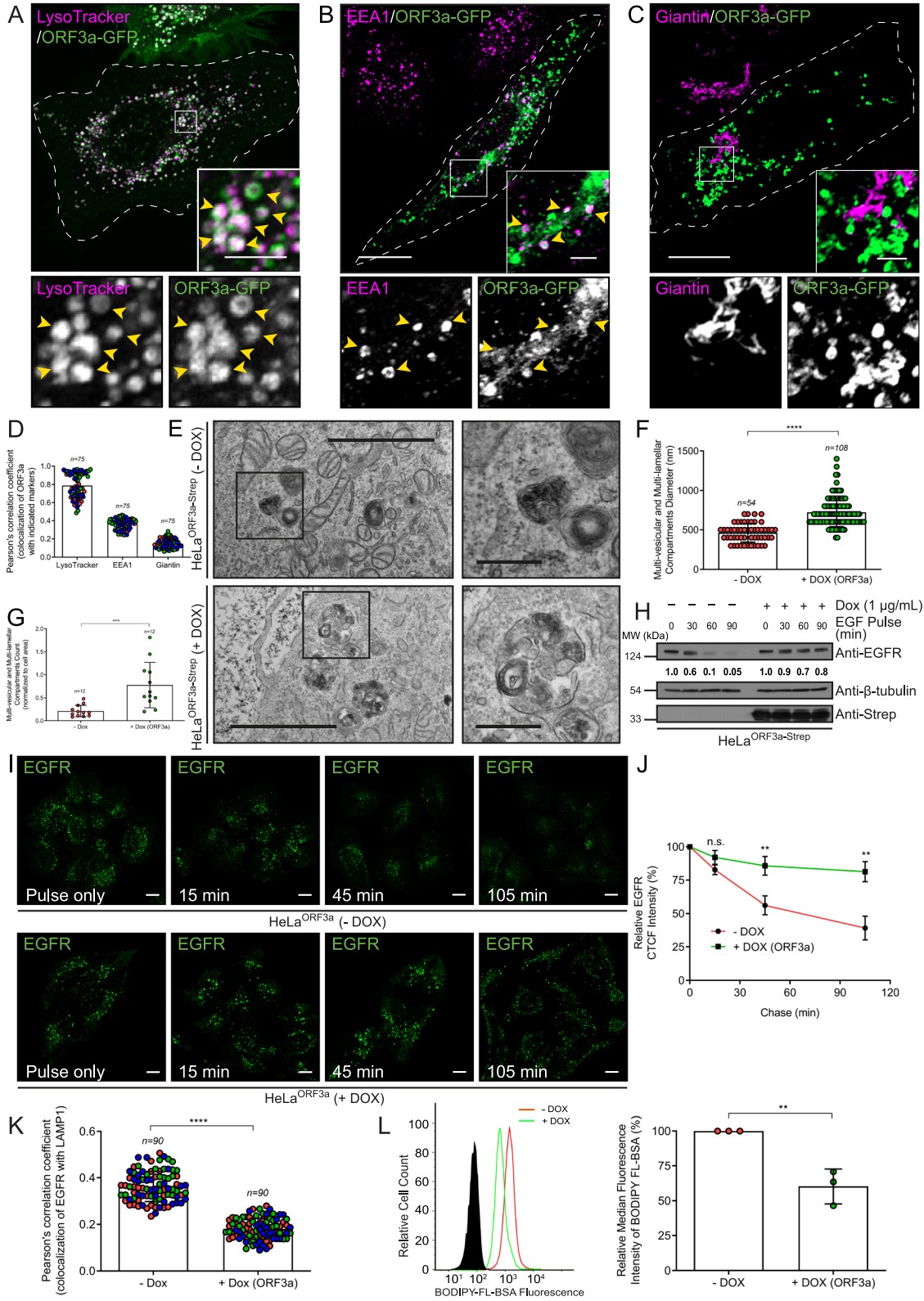

ultracentrifugation[22], we confirmed the localization of ORF3a in lysosomal fractions. The mitochondrial marker TOM20 and peroxisome marker catalase were mostly absent, whereas the lysosomal marker LAMP1 was enriched in these fractions, confirming the efficiency of fractionation (Supplementary Fig. S1I). We also confirmed the lysosomal localization of ORF3a using the LYSO-IP approach[23], in which ORF3a was transiently transfected into HEK293T cells stably expressing the epitope-tagged lysosomal transmembrane protein, TMEM-192. We confirmed the LYSO-IP results by immunoblotting for the lysosomal markers LAMP1 and Arl8b. The peroxisomal marker, catalase, was used to confirm the purity of the lysosomal fraction (Supplementary Fig. S1J).

Notably, ORF3a expression led to vacuolation of lysosomes and the appearance of ring-like LAMP1-positive vesicles. Furthermore, lysosomal positioning was also altered in ORF3a-expressing cells, which lacked a peripheral lysosomal population (see insets in

**Fig. 1 | ORF3a localizes to lysosomes and inhibits endocytic cargo degradation.**
**A–C** Confocal images of HeLa cells expressing ORF3a-GFP stained for the indicated organelle markers. Arrowheads in the insets indicate colocalized pixels. Scale bars: 10 μm (main); 2 μm (inset). **D** Pearson's colocalization coefficient (PCC) quantification of ORF3a-GFP with the indicated markers, n = 75 cells examined over three independent experiments. **E** Representative TEM images of untreated and Dox-treated HeLa[ORF3a-Strep] cells. The inset indicates multivesicular and multilamellar compartments. Scale bars: 2 μm (main); 0.5 μm (inset). Quantification of the size (**F**) and number (**G**) of multi-vesicular and multi-lamellar compartments in untreated and Dox-treated HeLa[ORF3a-Strep] cells using TEM images, p < 0.0001 (**F**) and p = 0.0009 (**G**). Note: n in (**G**) represents the number of multi-vesicular and multi-lamellar compartments analyzed. **H** Untreated and Dox-treated HeLa[ORF3a-Strep] cells were serum-starved and pulsed with EGF for the indicated time periods. Cell lysates were immunoblotted (IB) for proteins indicated, n = 3. The numbers represent densitometric analysis of the EGFR band intensity normalized to the β-tubulin.

**I–K** Confocal micrographs of untreated, and Dox-treated HeLa[ORF3a-Strep] cells treated with EGF for 15 min (pulse only) and chased for the indicated time periods, followed by immunostaining with an anti-EGFR antibody. Scale bars: 10 μm. The degradation of EGFR (**J**) was evaluated using confocal micrographs by normalizing the residual mean EGFR fluorescence intensity at pulse only, 15 min, 45 min and 105 min time points to the mean EGFR fluorescence intensity in the pulse-only sample. For -DOX, n = 109, 92, 100 and 116 cells and for +DOX, n = 100, 100, 98 and 100 cells, respectively examined over three independent experiments, p = 0.103 (15 min), 0.0067 (45 min), 0.0033 (105 min). **K** PCC quantification of EGFR with LAMP1 was calculated, n = 90 cells examined over three independent experiments, p < 0.0001. **L** Representative histogram of the dequenched BODIPY-FL-BSA signal in untreated and Dox-treated HeLa[ORF3a-Strep] cells as analyzed by flow cytometry; the graph represents the relative percentage of Mean Fluorescence Intensity for dequenched BODIPY-FL-BSA calculated, n = 3, p = 0.0052. Quantified results are presented as mean ± S.D. using unpaired two-tailed Student's t test.

Supplementary Fig. S1K). These observations were corroborated by quantifying the average lysosome diameter and fractional distance, i.e., the mean distance of lysosomes relative to the maximum distance from the center of the nucleus to the cell periphery, in control and ORF3a-expressing cells (Supplementary Figs. S1L and S1M). Next, we performed transmission electron microscopy (TEM) imaging of thin sections of control and ORF3a-expressing cells and visualized multi-vesicular/multilamellar compartments, which are typical morphologies of late endocytic compartments. Notably, compartments containing intraluminal vesicles (ILVs) and other cargo membranes were strikingly enlarged and increased in number in the cells expressing ORF3a (Fig. 1E–G). These observations indicate that the majority of enlarged LAMP1/lysotracker-positive vesicles in ORF3a-expressing cells are compartments with numerous ILVs and heterogeneous cargo membranes.

The SARS-CoV-2 ORF3a contains a YXXΦ sorting motif (160–163 amino acids) and a double glycine motif (187–188 amino acids) that are required for its intracellular transport from the Golgi apparatus to endosomes and lysosomes[18]. To investigate whether ORF3a localization to LEs/lysosomes is required for ORF3a-mediated lysosomal perturbations, we disrupted the YXXΦ sorting motif (ORF3a Y160A/V163G) and investigated the morphology and function of the lysosomes in these cells (Supplementary Fig. S1N). We confirmed that mutation of the YXXΦ sorting motif blocks ORF3a export from the Golgi apparatus (Supplementary Fig. S1O) as previously reported[18]. In contrast to that of ORF3a (WT), the expression of the ORF3a (Y160A/V163G) mutant did not cause enlargement of LAMP1-positive endosomes (Supplementary Fig. S1P). These observations were corroborated by quantifying the average diameter of LAMP1 puncta in cells expressing the ORF3a (WT) or ORF3a (Y160A/V163G) mutant (Supplementary Fig. S1Q).

## ORF3a blocks endocytic and autophagic cargo degradation in lysosomes

To directly assess lysosomal function in ORF3a-expressing cells, we analyzed the degradation of EGF-stimulated EGFR, a model cargo for lysosomal degradation[24]. To this end, we incubated control and ORF3a-expressing cells with EGF for increasing durations and measured the remaining EGFR levels in the cell lysates. As shown in Fig. 1H, in control cells, EGFR was significantly degraded after 60 and 90 min of incubation with EGF; in contrast, EGFR degradation was almost completely inhibited in ORF3a-expressing cells. To corroborate the observed defect in EGFR degradation, we quantified EGFR vesicular intensity in control and ORF3a-expressing cells incubated with EGF and chased for different durations using confocal microscopy. Indeed, the EGFR vesicular intensity remained relatively unchanged over the course of chase in ORF3a-expressing cells, whereas an ~2-fold decrease in EGFR vesicular intensity was observed in control cells after 45 min (Fig. 1I, J). Importantly, the colocalization of EGFR and LAMP1 was significantly lower in ORF3a-expressing cells than in control cells, indicating that

the fusion of EGFR-labeled vesicles with LAMP1-positive compartments was delayed upon ORF3a expression (see quantification of the PCC shown in Fig. 1K). We also analyzed lysosomal function in control and ORF3a-expressing cells by incubating cells with BODIPY-FL-BSA, an endocytic probe that fluoresces upon proteolytic cleavage in lysosomes[25,26]. ORF3a expression resulted in a 30–40% decrease in the BODIPY-FL-BSA signal, indicating that endocytic cargo delivery to lysosomes was delayed (Fig. 1L). Consistent with this, we found that the fusion of endocytic vesicles (labeled with dextran-568) with lysosomes (labeled with dextran-488) was significantly reduced in ORF3a-expressing cells (Supplementary Figs. S2A and S2B), supporting that ORF3a expression blocks late endosome (LE)–lysosome fusion.

Previous studies have shown that ORF3a inhibits the fusion of autophagosomes and lysosomes by disrupting the assembly of the vesicle fusion machinery[16,20,27]. We also assessed the effect of ORF3a on autophagic flux by measuring the amount of lipidated LC3 (LC3b-II) in cells with increasing levels of ORF3a. As shown in Supplementary Fig. S2C, there was an ~5.7-fold accumulation of LC3b-II within 24 h of ORF3a induction. Like that of LC3b-II, we observed ~3-fold accumulation of the autophagy substrate p62 within 24 h of ORF3a induction (Supplementary Fig. S2C). We corroborated these observations by quantifying LC3b and p62 puncta, which showed an average increase of ~5-fold and ~4.5-fold, respectively, in LC3b and p62 puncta per cell after 24 h of ORF3a expression (Supplementary Fig. S2D–G). A very insignificant increase in LC3b-II and p62 was observed in ORF3a-expressing cells upon treatment with bafilomycin A1 (BafA1), an inhibitor of lysosomal acidification, demonstrating that autophagic flux was already blocked upon ORF3a expression; thus, no additional increase was observed upon disruption of lysosomal function (Supplementary Figs. S2H, S2I). Thus, in line with the findings of previous reports, we also found that ORF3a does not inhibit autophagosome formation but rather blocks the fusion of autophagosomes with lysosomes. To visualize autophagosome-lysosome fusion, we measured the colocalization of LC3b and LAMP1 in serum-starved and BafA1-treated (to ensure maximum autolysosome formation) cells. As expected, in cells expressing ORF3a, LC3b/LAMP1 colocalization decreased dramatically (Supplementary Figs. S2J, S2K, and the quantification of the PCC is shown in Supplementary Fig. S2L).

Next, to investigate whether ORF3a lysosomal localization is required for ORF3a-mediated blockage of endocytic and autophagic cargo degradation, we generated tetracycline-inducible stable HeLa cell lines expressing HA-tagged ORF3a wild-type (HeLa[ORF3a (WT)]) or the Y160A/V163G mutant (HeLa[ORF3a (Y160A/V163G)]). We analyzed EGF-stimulated EGFR degradation in the two cell lines and found that, unlike ORF3a (WT) expression, which leads to a dramatic block in EGFR degradation, ORF3a (Y160A/V163G) mutant expression had no significant effect on EGFR degradation compared to that in the control (-Dox) (Supplementary Fig. S2M). Similarly, the expression of ORF3a (Y160A/V163G) mutant did not alter autophagic flux, as observed by

the levels of the autophagic protein LC3b-II and the autophagy substrate p62, indicating that the lysosomal localization of ORF3a is required for its ability to block lysosomal function (Supplementary Fig. S2N).

As one of the hallmark features of lysosomal dysfunction is an alteration in lysosomal pH, we next investigated whether ORF3a expression alters lysosomal pH. To this end, we employed the Lyso-Sensor Yellow/Blue DND-160, a pH-sensitive dye that allows ratiometric measurement of intraorganellar pH (ref. [61]). Notably, we found that while the average pH of control cells (-DOX) was $5.65 \pm 0.13$, the pH increased to $6.21 \pm 0.13$ in ORF3a (WT)-expressing cells. In contrast, in cells stably expressing the ORF3a (Y160A/V163G) mutant, the lysosomal pH was not altered ($5.32 \pm 0.22$) and remained similar to that of control cells ($5.54 \pm 0.13$) (Supplementary Fig. S2O). Overall, we found that ORF3a blocks endocytic and autophagic cargo-vesicle fusion with lysosomes, results in vacuolation of lysosomal compartments, and disrupts the lysosomal pH.

## ORF3a inhibits the interaction of the HOPS subunit Vps39 with the multivalent adaptor protein PLEKHM1

ORF3a has previously been shown to directly bind to Vps39, one of the six subunits of the multisubunit tethering factor HOPS complex. ORF3a expression impaired the interaction of the HOPS complex with syntaxin17 (STX17) and inhibited the assembly of the STX17-SNAP29-VAMP8 trans-SNARE complex, which mediates autophagosome-lysosome fusion[16,28]. We also confirmed that ORF3a interacts with Vps39 and other subunits of the HOPS complex, although the interaction was most prominent with Vps11 (which directly binds to Vps39) and Vps18 and weaker with Vps33a and Vps41 (Fig. 2A, B). Notably, Vps39 also interacts with PLEKHM1, a known effector of the small GTP-binding protein Rab7, which interacts with the small G protein Arl8b to orchestrate the HOPS-dependent fusion of late endosomes and autophagosomes with lysosomes[21,29]. We sought to test whether ORF3a alters Vps39 and PLEKHM1 interactions. Interestingly, as shown in Fig. 2C–E, Vps39 co-immunoprecipitated with PLEKHM1, and conversely, PLEKHM1 with Vps39 was dramatically reduced in the presence of ORF3a, which was co-immunoprecipitated with Vps39 and PLEKHM1. The interaction of PLEKHM1 with Vps11, which directly interacts with Vps39 as part of the HOPS complex, was also reduced in ORF3a-expressing cells. We found little or no change in the interaction of PLEKHM1 with other HOPS subunits, Vps41, Vps18, or Vps33a, in ORF3a-expressing cells (Fig. 2F). These findings suggest that ORF3a forms two distinct complexes in the cell: one where it interacts with Vps39 and Vps11 and another where it interacts with PLEKHM1 and subunits of the HOPS complex, except Vps39.

To better understand how ORF3a affects PLEKHM1 function, we visualized the localization of endogenous PLEKHM1 in ORF3a-expressing cells. Surprisingly, the number of PLEKHM1 puncta dramatically increased in ORF3a-expressing cells (Fig. 2G, and the quantification is shown in Fig. 2H). As expected from its endolysosomal localization, ORF3a colocalized with Rab7 on these enlarged puncta (see insets in Fig. 2G). Consistent with the increase in colocalization, co-immunoprecipitation of PLEKHM1 with Rab7 increased in ORF3a-expressing cells. Notably, ORF3a co-immunoprecipitated with Rab7, indicating that ORF3a forms a complex with Rab7 and PLEKHM1 (Fig. 2I, J). We noted that the membrane localization of Rab7, as indicated by the number and diameter of the Rab7 puncta, was significantly greater in the ORF3a-expressing cells than in the control cells (Fig. 2K, L). Consistent with these findings, we observed increased levels of Rab7 and PLEKHM1 in membrane fractions in the presence of ORF3a (Supplementary Fig. S3A). Furthermore, the levels of Rab7, but not LAMP1, was increased in lysosomal fractions isolated by LYSO-IP upon ORF3a expression (Supplementary Fig. S3B). Taken together, our results suggest that by binding to Vps39, ORF3a disrupts the interaction between Vps39 and PLEKHM1 and enhances the interaction

between PLEKHM1 and Rab7 on late endosomes, wherein ORF3a is also part of this latter complex.

## ORF3a expression promotes Rab7 activation

Our findings thus far show that ORF3a expression leads to enhanced membrane localization of the late endosomal small G protein Rab7 and increased association of Rab7 with PLEKHM1. Similar to other Rab effectors that stabilize Rab proteins in their active GTP-bound state upon binding[30], PLEKHM1 has been shown to promote Rab7 activation to its GTP-bound state[29,31,32]. As the presence of ORF3a led to an increase in the PLEKHM1 and Rab7 complex and enhanced Rab7 membrane localization (indicative of Rab activation), we hypothesized that ORF3a promotes Rab7 activation. To test this hypothesis, we employed a fluorescence recovery after photobleaching (FRAP) assay to determine the dynamics of Rab7 between the membrane and the cytosol upon ORF3a expression (Fig. 3A, B, and Supplementary Movie 1). We found that, on average, ~55% of the GFP-Rab7 signal was present in the mobile fraction in control cells, and the average $t_{1/2}$ was ~9 s (Fig. 3C, D). In ORF3a-expressing cells, we found significantly delayed Rab7 recovery on membranes post-bleaching, with an average of ~33% of the GFP-Rab7 signal in the mobile fraction and an average $t_{1/2}$ of ~13 s (Fig. 3C, D). Thus, ORF3a expression reduced the mobility of Rab7 between the membrane and the cytosol, suggesting that ORF3a expression impaired the dynamic shuttling of Rab7 between the GDP and GTP-bound states. To directly assess whether the levels of active or GTP-bound Rab7 in ORF3a-expressing cells were increased, we incubated control and ORF3a-expressing cell lysates with a GST-RILP-Rab7 binding domain (mR7BD) fragment that specifically binds to Rab7 in its GTP-bound form[33]. Compared with control cells, ORF3a-expressing cells showed a striking increase in GTP-bound Rab7 levels (Fig. 3E, F). An increase in active or GTP-bound Rab7 also leads to enhanced interaction of Rab7 with its effectors, as Rab GTPases interact with their effectors in a GTP-bound state. To this end, we analyzed the interaction of Rab7 with its known effectors RILP and ORP1L[34,35] in control and ORF3a-expressing cells. Indeed, we found that co-immunoprecipitation of RILP and ORP1L with Rab7 was strongly enhanced in ORF3a-expressing cells, and ORF3a was present in this complex (Fig. 3G and Supplementary Fig. S3C), confirming that ORF3a promotes Rab7 activation. In contrast to the WT, the ORF3a mutant, which is unable to export out of the Golgi (ORF3a Y160A/V163G), had no significant effect on the size or number of Rab7 puncta or on the level of GTP-bound Rab7 in cell lysates, indicating that the presence of ORF3a on LEs/lysosomes is required for its ability to promote Rab7 activation (Supplementary Fig. S3D–G).

Next, we assessed whether the natural variants of ORF3a present in different SARS-CoV-2 variants could promote Rab7 activation. According to the GISAID database (repository of COVID-19 sequences), the natural mutations found in ORF3a are Q57H/S171L (beta variant), S253P (gamma variant), S26L (delta variant), and T223I (omicron variant) in SARS-CoV-2 variants (Supplementary Fig. S3H). To determine whether these mutations change the function of ORF3a compared to ORF3a (WT), we constructed different mutants of ORF3a and expressed them in HeLa cells. We found that all the natural ORF3a variants localized to LEs/lysosomes and caused the enlargement of LAMP1-positive endosomes, similar to what was observed for ORF3a (WT) (Fig. 3H and Supplementary S3I). Next, we tested the interaction with Vps39 and found that all the natural variants of ORF3a interact with Vps39. One of the natural variants, ORF3a (Q57H/S171L), did not co-immunoprecipitate with Vps39 as much as did ORF3a (WT) (Supplementary Fig. S3J). Notably, residue S171 of ORF3a has been shown to be crucial for binding to Vps39, and mutation of S171 to glutamic acid (E) disrupts binding to Vps39[17]. Finally, all the natural variants of ORF3a increased the size and number of Rab7 puncta as well as the amount of GTP-bound Rab7 in cell lysates, similar to what was observed for ORF3a (WT) (Fig. 3H–J and the quantification is shown in Supplementary

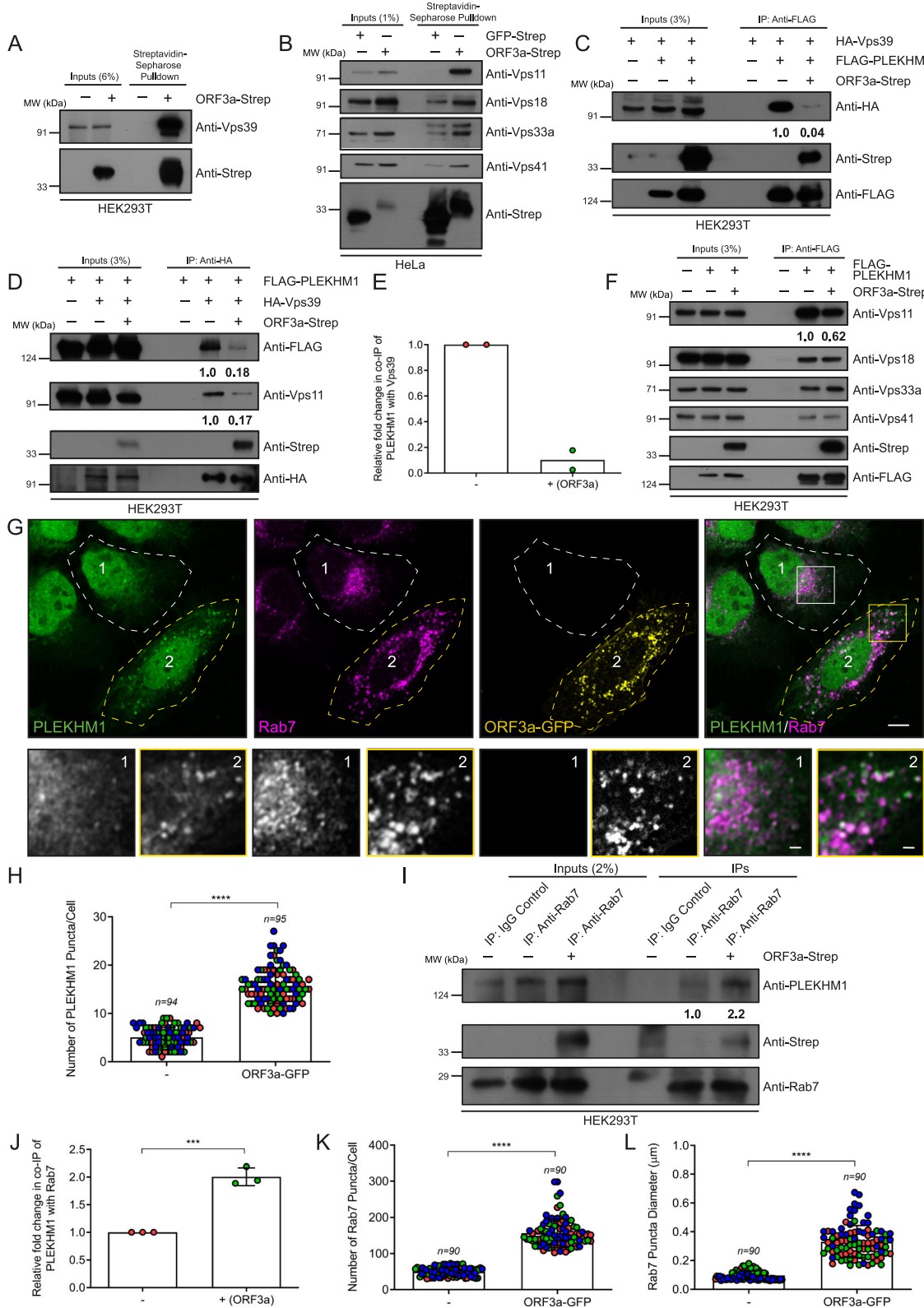

Figs. S3K, S3L). These results suggest that the localization and function of ORF3a in lysosomes are likely conserved during the evolution of SARS-CoV-2 variants.

## SARS-CoV-2 infection promotes Rab7 activation in an ORF3a-dependent manner

Next, we determined whether increased Rab7 activation was also observed in cells infected with SARS-CoV-2. To this end, we visualized the localization of endogenous Rab7 in SARS-CoV-2-infected Vero E6 cells. As shown in Supplementary Fig. S4A, cytosolic or diffuse Rab7 staining was strikingly reduced in infected cells (positive for the N antigen of SARS-CoV-2), and Rab7-positive endosomes were strongly clustered in the perinuclear region. Moreover, the average Rab7 puncta area per cell was significantly greater in SARS-CoV-2-infected cells than in control cells (Supplementary Fig. S4B). Next, we collected lysates from naturally susceptible Vero E6 cells infected with SARS-

**Fig. 2 | ORF3a binds to the HOPS subunit Vps39 and inhibits the interaction between Vps39 and PLEKHM1. A** Untransfected and ORF3a-Strep-transfected HEK293T cell lysates were incubated with streptavidin-sepharose beads. The precipitates were IB with the indicated antibodies, n = 2. **B** GFP- and ORF3a-Strep-expressing HeLa cell lysates were incubated with streptavidin-sepharose beads. The precipitates were subjected to IB with the indicated antibodies, n = 2. **C** Lysates of HEK293T cells expressing the indicated proteins were immunoprecipitated (IP) with anti-FLAG antibodies conjugated beads and subjected to IB with the indicated antibodies, n = 2. The values represent the densitometric analysis of the HA-Vps39 band intensity normalized to the input and direct IP of FLAG-PLEKHM1. **D** Lysates of HEK293T cells expressing the indicated proteins were subjected to IP with anti-HA antibodies conjugated beads and IB with the indicated antibodies, n = 2. The values represent the densitometric analysis of co-immunoprecipitated FLAG-PLEKHM1 and Vps11. **E** Densitometric analysis of co-immunoprecipitated FLAG-PLEKHM1. **F** Lysates of HEK293T cells expressing the indicated proteins were subjected to IP with anti-FLAG antibodies conjugated beads and IB with the indicated antibodies,

n = 2. The values represent the densitometric analysis of co-immunoprecipitated Vps11 band intensity. **G** Representative confocal images of HeLa cells transfected with the ORF3a-GFP-expressing plasmid and immunostained for endogenous Rab7 and PLEKHM1, n = 3. Insets 1 and 2 indicate the untransfected and ORF3a-GFP-transfected cells, respectively. Scale bars: 10 μm (main); 2 μm (inset). **H** Quantification of the number of PLEKHM1 puncta in wild-type (untransfected) and ORF3a-GFP-expressing HeLa cells, n = 94 or 95 cells examined over three independent experiments, as indicated, p < 0.0001. **I** Lysates of HEK293T cells expressing the indicated proteins were subjected to IP with anti-Rab7 antibodies conjugated beads and IB with the indicated antibodies, n = 3. The values in (**I**) and graph (**J**) represent the densitometric analysis of co-immunoprecipitated endogenous PLEKHM1, n = 3, p = 0.0004. **K, L** Quantification of Rab7 puncta in wild-type (untransfected) and ORF3a-GFP-expressing HeLa cells, n = 90 cells examined over three independent experiments, p < 0.0001. Quantified results are presented as mean ± S.D. using unpaired two-tailed Student's t test.

CoV-2 for 48 h and incubated them with GST-RILP-mR7BD to pull down and assess active Rab7 upon SARS-CoV-2 infection. We found a striking increase in GTP-bound Rab7 levels in SARS-CoV-2-infected cells compared to those in uninfected cells (Fig. 4A, B). These findings were corroborated in human adenocarcinoma A549 cells expressing ACE2 (Supplementary Fig. S4C). Like in Vero E6 cells, we observed enhanced Rab7 activation upon SARS-CoV-2 infection (Supplementary Fig. S4D).

To investigate whether Rab7 activation in SARS-CoV-2-infected cells is dependent on ORF3a, we depleted ORF3a from SARS-CoV-2-infected cells using siRNA and assessed the status of Rab7 activation. The efficiency of ORF3a silencing was ~70%, as confirmed by western blotting (Supplementary Fig. S1A). We first assessed whether this level of ORF3a knockdown was sufficient to observe the functional phenotype of virus-infected cells. We used two parameters to assess ORF3a function: (1) SARS-CoV-2 replication and (2) autophagic flux. As shown in Fig. 4C–F, SARS-CoV-2 replication was significantly reduced in ORF3a-depleted cells, as determined by qRT-PCR analysis of the expression of the *E* and *orf1ab* SARS-CoV-2 genes (Fig. 4E), as well as by measuring the intracellular (in cellular lysates) (bottom panels in Fig. 4C and quantification shown in Fig. 4D) and extracellular (in culture media) protein levels of the SARS-CoV-2 N-antigen (Fig. 4F). Furthermore, as observed by the protein levels of the autophagosomal marker protein LC3b and the autophagy substrate p62, ORF3a depletion reduced the impaired autophagic flux phenotype observed in SARS-CoV-2-infected cells (top and middle panels in Fig. 4C).

To test whether Rab7 hyperactivation in SARS-CoV-2-infected cells is ORF3a-dependent, we analyzed Rab7 localization and activation status in ORF3a-depleted cells. Indeed, ORF3a knockdown during SARS-CoV-2 infection restored the size and distribution of Rab7-positive endosomes to levels similar to those was observed in uninfected cells (Fig. 4G, H). Consistent with the role of ORF3a in Rab7 activation during SARS-CoV-2 infection, we found that ORF3a depletion reduced the level of GTP-bound Rab7 in SARS-CoV-2-infected cells to a level similar to that observed in uninfected cells (Fig. 4I, J). As expected, viral replication, as measured by immunoblotting for the N-antigen and spike (S) protein, revealed a reduced viral load in ORF3a-depleted cells (Fig. 4I). Taken together, our findings strongly support that the virulence factor ORF3a mediates the hyperactivation of Rab7 in cells infected with SARS-CoV-2.

### ORF3a impairs the interaction of Rab7 with TBC1D5 in a Vps39-dependent manner

A previous study reported that the interaction of ORF3a with Vps39 is required for ability to inhibit autophagic flux and promote lysosomal exocytosis[17]. We investigated whether the ORF3a-Vps39 interaction is required for ORF3a-mediated Rab7 activation. To this end, we created point mutations in ORF3a, namely, S171E and W193R, which were previously shown to disrupt ORF3a binding to Vps39 (Supplementary

Fig. S4E). Interestingly, in contrast to that of ORF3a (WT), the expression of the ORF3a (S171E) or ORF3a (W193R) mutant failed to cause a change in Rab7 localization or Rab7 activation status, suggesting that ORF3a binding to Vps39 is required for its downstream role in enhancing active Rab7 levels (Fig. 5A–D and Supplementary Fig. S4F). We also tested whether the Q57E and S58L/Q116L mutations in ORF3a (which reduce ORF3a-mediated Ca²⁺ permeability) have any effect on ORF3a-mediated Rab7 activation. Notably, similar to those of ORF3a (WT), both the Q57E and S58L/Q116L mutants exhibited increases in the size and number of Rab7 puncta (Fig. 5A, B, and Supplementary Fig. S4F). These results indicate that ORF3a-mediated Rab7 activation requires Vps39 binding but is independent of its putative role in mediating Ca²⁺ permeability.

To confirm that Vps39 binding is essential for ORF3a function, we rescued the effect of ORF3a depletion in SARS-CoV-2-infected cells using ORF3a (WT) or Vps39 binding-defective (S171E and W193R) mutants. We assessed GTP-bound Rab7 levels in SARS-CoV-2-infected Vero E6 cell lysates depleted of ORF3a and expressing either siRNA-resistant ORF3a (WT) or the S171E or W193R mutant. As shown in Fig. 5E, F, while ORF3a (WT) efficiently restored Rab7-GTP levels, both Vps39 binding-defective mutants were unable to rescue the effect of ORF3a depletion. We observed a similar result in the rescue of viral replication by ORF3a (WT) but not by Vps39-binding-defective ORF3a mutants, as assessed by measuring the total protein levels of the N-antigen (Fig. 5E) and quantifying of the expression levels of the SARS-CoV-2 *E* and *orf1ab* genes (Fig. 5G). In line with our conclusion that Vps39 binding is required for ORF3a mediated viral replication, we observed reduced total protein levels of the N-antigen and reduced expression of the SARS-CoV-2 *E* and *orf1ab* genes in Vps39 siRNA-treated SARS-CoV-2-infected cells compared to those in control cells (Supplementary Fig. S4G–I). Thus, we conclude that ORF3a binding to Vps39 is essential for its role in mediating Rab7 hyperactivation and viral replication during SARS-CoV-2 infection.

To explore the mechanism by which ORF3a leads to sustained Rab7 activation, we investigated whether the ORF3a-Vps39 subcomplex affects the association of Rab7 with its known GTPase activating proteins (GAPs), TBC1D5 and TBC1D15, which mediate GTP hydrolysis of Rab7 on late endosomes[36–39]. We found that Vps39 interacts with TBC1D5 but not with TBC1D15 (Supplementary Figs. S4J, S4K). Moreover, co-immunoprecipitates of ORF3a (WT) but not Vps39 binding-defective mutants, i.e., ORF3a (S171E) and (W193R), with TBC1D5 suggested that TBC1D5 interacts with ORF3a in a Vps39-dependent manner (Fig. 5H). We also noted that upon expression of ORF3a (WT), the interaction of Vps39 with TBC1D5 increased, whereas no such increase was observed upon expression of the ORF3a (S171E) or (W193R) mutant (Supplementary Fig. S4L). These results indicate that the ORF3a and Vps39 complex is conducive to binding TBC1D5, but not the Rab7-PLEKHM1 complex (Fig. 2C–E and I).

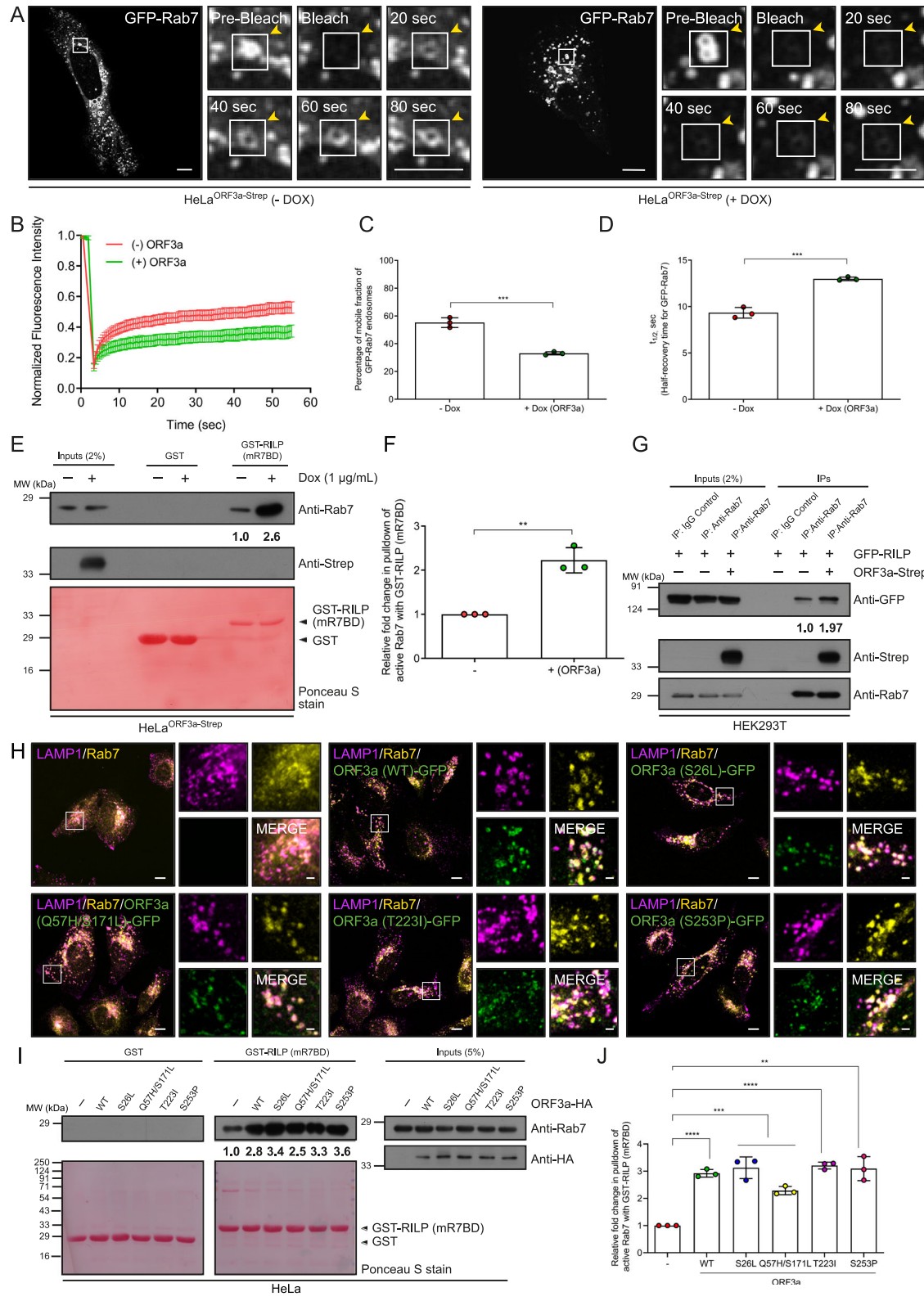

To investigate whether ORF3a regulates the interaction of TBC1D5 with Rab7, we performed co-immunoprecipitation of TBC1D5 with Rab7 in the presence of ORF3a. ORF3a expression led to a decrease in the interaction of TBC1D5 with Rab7 when the interaction was analyzed under endogenous conditions (Fig. 5I). Importantly, unlike that of ORF3a (WT), the expression of Vps39-binding defective mutants of ORF3a (S171E or W193R) did not significantly decrease the interaction of Rab7 with TBC1D5 (Fig. 5J), which is in agreement with our previous

observation that these ORF3a mutants do not affect Rab7 activation (Fig. 5C, D). Notably, no change in the interaction of Rab7 with TBC1D15 was observed in the presence of ORF3a, indicating that this effect was specific to TBC1D5 (Supplementary Fig. S4M). Next, we investigated whether the interaction of Rab7 with GAP was altered upon SARS-CoV-2 infection. Notably, the interaction of Rab7 with TBC1D5 significantly decreased upon SARS-CoV-2 infection (Fig. 5K, L). Taken together, our results suggest that the SARS-CoV-2 virulence

**Fig. 3 | Expression of ORF3a or its natural variants promotes Rab7 activation.**
**A** Representative time-lapse images of GFP-Rab7 endosomes before (pre-bleach) and after (bleach) photobleaching in untreated and Dox-treated HeLa[ORF3a-Strep] cells; see Supplementary Movie 1. **B** FRAP recovery curves of GFP-Rab7 endosomes in untreated and Dox-treated HeLa[ORF3a-Strep] cells. The vertical-axis denotes the normalized fluorescence intensity of GFP-Rab7, corrected for fluorescence decay and background fluorescence. Data points are plotted as the mean ± S.E.M, $n = 20$ (-DOX) and 19 ( + DOX) cells examined over three independent experiments. The percentage of the mobile fraction (**C**) and half-time ($t_{1/2}$) of recovery (**D**) for GFP-Rab7 signal were calculated, $n = 3$, $p = 0.0005$ (**C**) and 0.0004 (**D**). **E** GST and GST-RILP proteins immobilized on beads were incubated with lysates of untreated and Dox-treated HeLa[ORF3a-Strep] cells. The precipitates were IB with anti-Rab7 antibodies, $n = 3$. The values written in (**E**) and graph (**F**) represent densitometric analysis for the levels of active Rab7 pulldown, $n = 3$, $p = 0.0018$. **G** Lysates of HEK293T cells expressing the indicated proteins were IP with anti-Rab7 antibodies conjugated beads and IB with the indicated antibodies. The values represent the densitometric analysis of co-immunoprecipitated GFP-RILP, $n = 2$. **H** Representative confocal micrographs of HeLa cells wild-type or expressing ORF3a (WT)-GFP or different natural variants of ORF3a and immunostained for LAMP1 and Rab7. The insets show the increase in the size of LAMP1-positive endosomes and the increase in the membrane localization of Rab7 observed upon the expression of ORF3a (WT) or its natural variants, $n = 3$. **I** GST and GST-RILP proteins immobilized on beads were incubated with lysates of HeLa cells expressing indicated proteins. The precipitates were IB with anti-Rab7 antibodies. The values in (**I**) and graph (**J**) represent densitometric analysis of the levels of active Rab7 pulldown, $n = 3$, $p < 0.0001$ (WT and T223I), 0.0008 (S26L), 0.0001 (Q57H/S171L), 0.0012 (S253P). Quantified results are presented as mean ± S.D. using unpaired two-tailed Student's $t$ test. Scale bars: 10 μm (main); 2 μm (inset).

factor ORF3a inhibits the interaction of Rab7 with its GAP TBC1D5, leading to sustained GTP-bound Rab7 on late endosomes.

## The GTP-bound form of Rab7 is essential for SARS-CoV-2 infection

Thus far, the data indicate that the SARS-CoV-2 genome encodes the virulence factor ORF3a which binds to the Vps39 subunit of the HOPS complex and disrupts the Rab7 GTPase cycle, resulting in Rab7 hyperactivation in virus-infected or ORF3a-expressing cells. We next checked whether the presence of Rab7 in an active or constitutive GTP-bound state was beneficial for SARS-CoV-2 replication. First, we confirmed that Rab7 is required for viral replication, as depletion of Rab7 in SARS-CoV-2-infected cells reduced the total protein level of the N-antigen and reduced the expression of the SARS-CoV-2 *E* and *orf1ab* genes (Fig. 6A–C). Next, we investigated whether the defect in viral replication upon Rab7 depletion could be rescued by the WT, constitutively GTP-bound (Q67L), or constitutively GDP-bound (T22N) forms of Rab7. Surprisingly, we found a strong positive correlation between active Rab7 and SARS-CoV-2 viral replication, as measured by the expression of the *E* and *orf1ab* SARS-CoV-2 genes and by measuring the intracellular and extracellular protein levels of the SARS-CoV-2 N-antigen (Fig. 6D–F). Rescue of the viral load with constitutive GTP-bound Rab7 (Q67L) was significantly more efficient than that with Rab7 (WT), while expression of constitutive GDP-bound Rab7 (T22N) dramatically reduced the viral load above the effect of Rab7 gene knockdown (Fig. 6D–F). The reason for this difference could be the dominant-negative effect of Rab7 (T22N) on endogenous Rab7, which was still present due to partial gene knockdown.

To support these findings, we tested chemical compounds that are characterized as either Rab7 activators (ML-098)[40] or Rab7 inhibitors (CID-1067700)[41]. First, we tested whether these compounds could activate or inhibit Rab7 in Vero E6 cells. To this end, we incubated Vero E6 cell lysates treated with DMSO (control), ML-098, or CID-1067700 with GST-RILP (mR7BD) and immunoblotted for active levels of Rab7. As shown in Supplementary Fig. S4N, treatment with ML-098 or CID-1067700 increased or decreased the level of active Rab7, respectively, in the cells. Next, we treated SARS-CoV-2-infected Vero E6 cells with these compounds and analyzed the viral replication. In line with our results obtained using GTP-locked and GDP-locked forms of Rab7, we found a strong positive correlation between active levels of Rab7 and SARS-CoV-2 replication, as shown by the expression of *E* and *orf1ab* SARS-CoV-2 genes and by measuring the intracellular and extracellular protein levels of the SARS-CoV-2 N-antigen (Fig. 6G–J).

Next, we questioned whether the loss of ORF3a in SARS-CoV-2-infected cells is compensated by active Rab7. To this end, we expressed the constitutively GTP-locked (Q67L) form of Rab7 in ORF3a siRNA-treated SARS-CoV-2-infected cells. Indeed, we found significant rescue of SARS-CoV-2 replication when activated Rab7 was present in ORF3a-depleted cells (Fig. 6K–M). These findings suggest that a key role of ORF3a is to promote Rab7 hyperactivation, which in turn is required for viral replication.

## ORF3a blocks CI-M6PR recycling from Rab7-positive late endosomes and impairs the sorting of newly synthesized procathepsin D to lysosomes

Proper GTP-GDP cycling by Rab7 is crucial for late endosome positioning and fusion with lysosomes, as well as for recycling cargo from late endosomes to the trans-Golgi network (TGN) and/or plasma membrane[42]. One of the best-characterized cargo receptors that recycles from late endosomes back to the TGN in a Rab7-dependent manner is the cation-independent mannose 6-phosphate receptor (CI-M6PR), which mediates the sorting of mannose 6-phosphate-labeled inactive forms of lysosomal hydrolases from the TGN to late endosomes[43]. Indeed, recruitment of TBC1D5 by the retromer complex, which is a highly conserved heterotrimeric cargo recognition complex, is required to inactivate Rab7 and likely promote cargo recycling from late endosomes[44]. As ORF3a expression led to defects in lysosomal function and promoted Rab7 activation (Figs. 1–6), we next analyzed whether ORF3a-dependent Rab7 activation causes defects in CI-M6PR trafficking and consequently lysosomal hydrolase sorting. Strikingly, compared to those in control cells, where CI-M6PR was mostly distributed in the perinuclear region, the distribution of CI-M6PR was more punctate in the ORF3a-expressing cells (Fig. 7A). More importantly, CI-M6PR was stuck to enlarged Rab7 endosomes, as reflected by a dramatic increase in the colocalization of CI-M6PR and Rab7 in ORF3a-expressing cells (see insets in Fig. 7A and the quantification of PCC is shown in Fig. 7B). This enhanced colocalization of CI-M6PR with Rab7 suggested that CI-M6PR recycling from late endosomes is blocked by ORF3a. We also analyzed the colocalization of Rab7 with the retromer subunit Vps35, which is implicated in cargo retrieval from late endosomes back to the TGN[45]. In agreement with previous reports showing that Vps35 interacts with the GTP-bound form of Rab7[46], we found enhanced colocalization of Vps35 with Rab7 in ORF3a-expressing cells (see insets in Fig. 7C and the quantification of PCC is shown in Fig. 7D).

Delayed or blocked CI-M6PR recycling to the TGN leads to missorting of inactive forms of lysosomal hydrolases from the TGN. To analyze this phenomenon, we employed the Retention Using Selective Hooks (RUSH) approach[25] to investigate the trafficking of newly synthesized cathepsin Z in control and ORF3a-expressing cells. Before biotin addition, as expected, we observed ER localization of the cathepsin Z construct with a streptavidin tag that was fused to the ER retention signal KDEL (hook) in both control and ORF3a-expressing cells. With increasing time after the addition of biotin until 120 min, we observed increased colocalization of cathepsin Z and LAMP1-GFP in control cells, indicating the delivery of the newly synthesized cathepsin Z to LEs/lysosomes (Fig. 7E, and the quantification of the PCC is shown in Fig. 7F). However, in ORF3a-expressing cells, while cathepsin

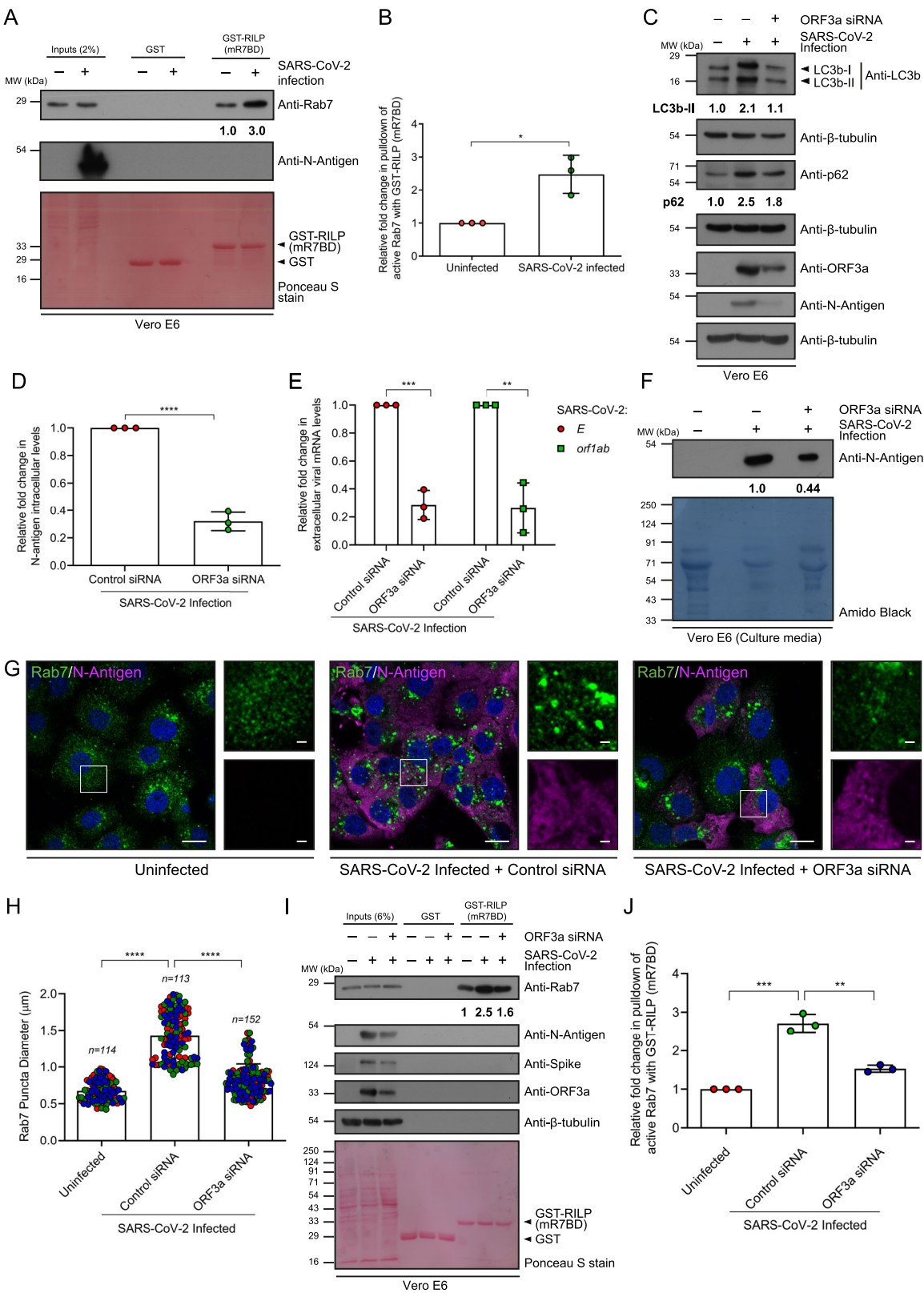

Z trafficking to the Golgi appeared to progress similarly to that in control cells, there was a significant delay in trafficking from the Golgi to lysosomes, as indicated by a reduced overlap with LAMP1-GFP at different time points after biotin addition (Fig. 7E, and quantification of the PCC is shown in Fig. 7F). Consistent with the defect in cathepsin Z sorting to lysosomes, we found a modest increase in pro-cathepsin D

levels in total cellular lysates, as well as in the extracellular media of ORF3a-expressing cells (Fig. 7G–I). We also analyzed cathepsin activity using an SiR-lysosome dye that fluoresces upon cleavage by cathepsin D[22]. Indeed, we found an ~2.4-fold decrease in the SiR-lysosome signal intensity in ORF3a-expressing cells (Fig. 7J, K). Taken together, our results show that ORF3a blocks CI-M6PR recycling from Rab7

**Fig. 4 | SARS-CoV-2 infection leads to Rab7 hyperactivation in an ORF3a-dependent manner. A** GST and GST-RILP proteins immobilized on beads were incubated with lysates from uninfected and SARS-CoV-2-infected Vero E6 cells. The precipitates were IB with indicated antibodies, *n = 3*. The values in (**A**) and graph (**B**) represent densitometric analysis of the levels of active Rab7 pulldown, *n = 3*, *p = 0.0116.* **C** Lysates of Vero E6 cells infected with SARS-CoV-2 and transfected with either control or ORF3a-siRNA were IB for the indicated proteins, *n = 3*. The values represent densitometric analysis of LC3b-II and p62 levels normalized to β-tubulin. **D** Relative fold-change in intracellular N-antigen levels in lysates of SARS-CoV-2-infected Vero E6 cells transfected with control or ORF3a-siRNA, *n = 3*, *p < 0.0001.* **E** Relative expression of SARS-CoV-2 *E* and *orf1ab* genes measured in the culture supernatants, *n = 3*, *p = 0.0003 (E gene), 0.002 (orf1ab gene).* **F** Protein levels of N-antigen in the culture supernatants. The values represent a densitometric analysis of the levels of extracellular N-antigen, *n = 1.* **G** Representative confocal images
of Vero E6 cells uninfected or infected with SARS-CoV-2 and transfected with either control or ORF3a-siRNA and immunostained for endogenous Rab7 and N-antigen. The insets show the differences in the sizes of the Rab7 puncta, *n = 3*. Scale bars: 10 μm (main); 2 μm (inset). **H** Quantification of Rab7 puncta size in uninfected and SARS-CoV-2-infected Vero E6 cells transfected with control or ORF3a-siRNA, *n = 114, 113 and 152 cells examined over three independent experiments, as indicated, p < 0.0001.* **I** GST and GST-RILP proteins immobilized on beads were incubated with lysates from uninfected and SARS-CoV-2-infected Vero E6 cells transfected with control or ORF3a-siRNA. The precipitates were IB with the indicated antibodies, *n = 3*. The values in (**I**) and graph (**J**) represent densitometric analysis of the levels of active Rab7 pulldown, *n = 3*, *p = 0.0002 (uninfected versus control siRNA-infected), 0.0012 (control versus ORF3a siRNA-infected).* Quantified results are presented as mean ± S.D. using unpaired two-tailed Student's *t* test.

endosomes to the TGN, which in turn impairs lysosomal hydrolase sorting from the TGN, eventually leading to extracellular secretion via the default pathway.

## ORF3a blocks the fusion of Rab7-positive and Arl8b-positive late endosomes and lysosomes

Previous studies have shown that Rab7-positive late endocytic vesicles transiently interact and frequently fuse with Arl8b-positive compartments, which are typically regarded as mature lysosomes[21,47]. Interestingly, these two small G proteins have opposing roles in regulating the motility and distribution of lysosomes, with Rab7 overexpression driving perinuclear clustering of lysosomes, whereas Arl8b overexpression promotes peripheral lysosome motility. Furthermore, Rab7 perinuclear compartments have also been shown to mature into Arl8b-positive compartments that eventually move to the cell periphery[47]. As ORF3a stalls Rab7 in its active GTP-bound state, we investigated how ORF3a expression affects the kiss and run events between Rab7- and Arl8b-containing endosomes. We observed that ORF3a expression (Fig. 8A), and SARS-CoV-2 infection (Fig. 8C) led to a reduced overlap between endogenous Rab7 and Arl8b endosomes, although the enlarged perinuclear Rab7-positive endosomes were also Arl8b-positive (the quantification of the PCC is shown in Fig. 8B, D). In fact, live-cell imaging revealed that when ORF3a was expressed, the number of fusion events between Rab7 and Arl8b endosomes was greatly reduced, even though both small G proteins were clustered around the perinuclear region (Fig. 8E, F, and Supplementary Movie 2).

To directly test whether ORF3a expression affects endosome fusion or maturation from Rab7 to Arl8b, we incubated cells with 10 nm paramagnetic particles (ferrofluid). Ferrofluid-containing endosomes mature from early to late compartments after endocytosis and eventually acquire lysosomal markers[48]. These maturation events can be identified by magnet-based separation of ferrofluid endosomes from cell homogenates at different time points, followed by immunoblotting for various endosomal marker proteins. Herein, we observed that the early to late endosome transitions marked by Rab5 and Rab7, respectively, were similar in the control and ORF3a-expressing cells. In contrast, the acquisition of Arl8b and LAMP1 on ferrofluid endosomes was strongly reduced in ORF3a-expressing cells compared to that in the controls (Fig. 8G). These data demonstrated that ORF3a impairs the transition or fusion of Rab7-positive late endosomes to Arl8b/LAMP1-positive endosomes.

Interestingly, we also noted that the mobile fraction and average speed of Arl8b-positive endosomes were modestly, but not significantly, increased in ORF3a-expressing cells than in control cells (Fig. 8H–K). In contrast, the mobile fraction and average speed of Rab7 compartments, which were primarily localized in the perinuclear region, were significantly reduced upon ORF3a expression (Fig. 8H, I, L, M). This finding is consistent with our previous results that ORF3a promotes the interaction of Rab7 with RILP, which in turn recruits the dynein-dynactin complex to late endosomes, resulting in perinuclear

accumulation (Fig. 3G and Supplementary Fig. S1M). We speculate that ORF3a expression disrupts efficient membrane exchange and fusion between these two small G protein compartments by stabilizing Rab7 in its GTP-bound state. This results in impaired degradation of endocytic and autophagic cargo, whereas the more dynamic and motile Arl8b-positive lysosomal compartments may represent the lysosomal population hijacked by coronaviruses for egress from infected cells, as demonstrated in a prior study[4].

## Discussion

Previous studies have established that ORF3a is a major accessory protein of SARS-CoV-2 that disrupts host lysosomal cargo degradation and promotes lysosomal exocytosis, which in turn facilitates viral egress[17,27]. A major interaction partner of ORF3a in host cells is Vps39, a subunit of the hexameric tethering factor complex known as the HOPS complex that mediates the tethering and fusion of late endosomes and autophagosomes with lysosomes[16,21,24,28]. ORF3a binding to Vps39 disrupts the interaction of the HOPS complex with the SNARE syntaxin 17 (STX17) and prevents the assembly of the STX17-SNAP29-VAMP8 SNARE complex, which is required for autophagosome-lysosome fusion. Thus, ORF3a binding to Vps39 blocks lysosomal function during autophagic cargo degradation[16,20]. ORF3a binding to Vps39 has also been shown to be necessary for lysosomal exocytosis[17], but how its interaction with Vps39 promotes lysosomal exocytosis has not been determined.

In this study, we found that ORF3a impairs the direct binding of Vps39 to PLEKHM1, a multidomain adaptor that also binds to the late endosomal/lysosomal small G proteins Rab7 and Arl8b, and the autophagosomal protein LC3 (Fig. 2C–E). PLEKHM1 and Arl8b recruit the HOPS complex to late endosomes and lysosomes, respectively, promoting late endosome/autophagosome -lysosome tethering and fusion[21,24,29,49]. Thus, ORF3a-mediated disruption of Vps39-PLEKHM1 binding abrogated the assembly of this fusion machinery, leading to defects in endosomal and autophagic cargo degradation by lysosomes (Figs. 1H–M, 2F and Supplementary Fig. S2).

Interestingly, ORF3a expression increased the association between PLEKHM1 and Rab7 (Fig. 2G–J). As the interaction of Rab7 with PLEKHM1 was previously reported to stabilize Rab7 in its GTP-bound state[29,31,32], we next assessed the status of Rab7 activation in ORF3a-expressing cells. Rab proteins function as molecular switches that alternate between an active GTP-bound state, in which they are reversibly recruited to target membranes, and an inactive GDP-bound state, in which they are predominantly cytosolic and dissociate from target membranes. In the GTP-bound state, Rab proteins recruit effectors that mediate downstream steps of vesicular transport, including cargo selection, vesicle budding, motility, tethering, and fusion with the target compartment[30]. Several independent experiments confirmed that ORF3a leads to an increase in GTP-bound Rab7 levels in cells. These changes included increased Rab7 levels in the lysosomal fractions (Supplementary Figs. S3A, S3B), reduced mobility

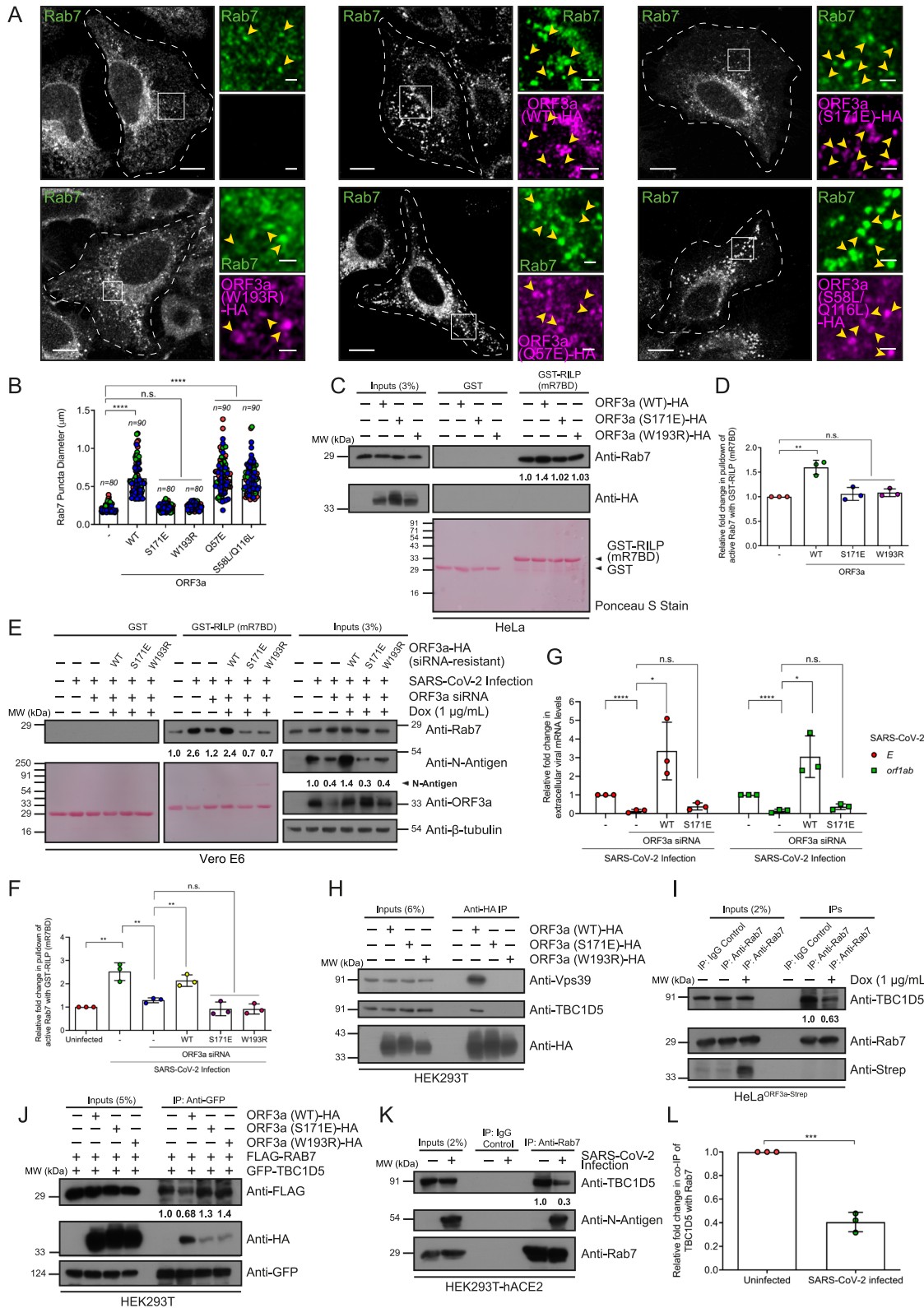

of Rab7 between the membrane and cytosol (Fig. 3A–D), enhanced pull-down of Rab7 with a RILP-Rab-binding domain fragment that recognizes the GTP-bound Rab7 form (Fig. 3E, F), and enhanced co-immunoprecipitation of the Rab7 effectors RILP and ORP1L with Rab7 (Fig. 3G, and Supplementary Fig. S3C). ORF3a LE/lysosome localization was required for its ability to mediate Rab7 hyperactivation, as a mutation in the YXXΦ sorting motif (160–163 amino acids) in the ORF3a cytoplasmic tail (which stalled ORF3a export from the TGN)

abrogated this phenotype of Rab7 hyperactivation (Supplementary Figs. S1N–Q and S3D–G).

Importantly, we confirmed that Rab7 was hyperactivated upon SARS-CoV-2 infection (Fig. 4A, B, and Supplementary Fig. S4A–D), suggesting that infection with the live virus recapitulates the ectopic expression of ORF3a. Furthermore, ORF3a depletion in SARS-CoV-2-infected cells and rescue with ORF3a (WT), but not with Vps39-binding defective mutants, confirmed that ORF3a is a virulence factor that

**Fig. 5 | ORF3a inhibits the interaction between Rab7 and TBC1D5 in a Vps39-dependent manner. A** Representative confocal micrographs of HeLa cells transfected with ORF3a (WT) or mutants and immunostained for the indicated proteins. Scale bars: 10 μm (main); 2 μm (inset). **B** Quantification of Rab7 puncta size in HeLa cells transfected with indicated ORF3a, n = 80 or 90 cells examined over three independent experiments, as indicated, p < 0.0001(WT, Q57E, S58L/Q116L), p = 0.6309(S171E), p = 0.0519(W193R). **C** GST and GST-RILP proteins immobilized on beads were incubated with HeLa cell lysates expressing the indicated proteins. The precipitates were IB with indicated antibodies. The values in (**C**) and graph (**D**) represent densitometric analysis of the levels of active Rab7 pulldown, n = 3, p = 0.002(WT), 0.4912(S171E), 0.1407(W193R). **E** GST and GST-RILP proteins immobilized on beads were incubated with lysates of uninfected and SARS-CoV-2-infected cells transfected with control or ORF3a-siRNA and expressing indicated proteins. The precipitates were IB with indicated antibodies. The values in (**E**) and graph (**F**) represent densitometric analysis of the levels of active Rab7 pulldown, n = 3, p = 0.0023(uninfected versus control siRNA-infected), 0.0058(control versus ORF3a siRNA-infected), 0.0056(ORF3a (WT) versus ORF3a siRNA), 0.1095(ORF3a

(S171E) versus ORF3a siRNA), 0.0555(ORF3a (W193R) versus ORF3a siRNA). **G** Relative expression of the *E* and *orf1ab* genes measured by qRT-PCR in the media, n = 3. For control versus ORF3a siRNA, p < 0.0001(*E* and *orf1ab* genes); for ORF3a (WT) versus ORF3a siRNA, p = 0.0228(*E*), 0.0108(*orf1ab*); for ORF3a (S171E) versus ORF3a siRNA, p = 0.1115(*E*), 0.0912(*orf1ab*). **H** Lysates of HEK293T cells expressing the indicated proteins were IP with anti-HA antibody-conjugated beads and IB with the indicated antibodies, n = 3. **I** Lysates of untreated and Dox-treated HeLa^ORF3a-Strep cells were IP using anti-Rab7 antibody-conjugated beads and IB with the indicated antibodies. The values represent densitometric analysis of co-immunoprecipitated TBC1D5, n = 3. **J** Lysates of HEK293T cells expressing the indicated proteins were IP with anti-GFP antibody-conjugated beads and IB with the indicated antibodies. The values represent the densitometric analysis of co-immunoprecipitated FLAG-Rab7, n = 3. **K** Lysates of uninfected and SARS-CoV-2-infected HEK293T-hACE2 cells were IP using anti-Rab7 antibody-conjugated beads and IB with the indicated antibodies, n = 3. The values in (**K**) and graph (**L**) represent the densitometric analysis of co-immunoprecipitated TBC1D5, n = 3, p = 0.0002.

maintains Rab7 activation and that Vps39 binding to ORF3a is required for this process (Fig. 5E–G). We found that all the natural variants of ORF3a present in different SARS-CoV-2 variants promoted Rab7 hyperactivation, suggesting that maintaining active Rab7 levels is essential for virus replication (Fig. 3H–J). In support of these findings, we found that ORF3a or Vps39 depletion greatly reduced the viral load during infection (Fig. 4C–F and Supplementary Fig. S4G–I). In addition, we found a strong correlation between Rab7 activation and virus replication, with viral production being significantly reduced upon Rab7 depletion. Rescue with the GTP-locked (Q67L) form of Rab7 resulted in enhanced virus production compared to rescue with Rab7 (WT), while the presence of the GDP-locked (T22N) form of Rab7 completely abrogated viral replication (Fig. 6A–F). Finally, we found that the expression of constitutively active Rab7 (Q67L) rescued the SARS-CoV-2 replication defect in ORF3a-depleted cells, suggesting that a crucial downstream function of ORF3a is to promote Rab7 activation, which in turn is required for viral replication (Fig. 6K–M).

One of the key cargo receptors that recycle from a maturing early/late endosome is CI-M6PR, which traffics back from endosomes to the TGN and captures M6P-modified precursor forms of lysosomal hydrolases for sorting toward late endosomes[43]. In ORF3a-expressing cells, the TGN distribution of CI-M6PR was greatly reduced, and CI-M6PR accumulated on Rab7-positive late endosomes (Fig. 7A, B). We also found that ORF3a-expressing cells exhibited increased recruitment of Vps35, a subunit of the retromer cargo selection complex, to Rab7-positive endosomes (Fig. 7C, D). Vps35 is recruited to maturing endosomes in part by binding to active Rab7[45,46], and stabilization of the Rab7 active state by ORF3a is expected to promote Vps35 recruitment to late endosomes. These findings suggest that hyperactivation of Rab7 prevents the release of the retromer complex and, in turn, prevents the recycling of membranes and membrane-bound cargo from late endosomes. These finding support the enlarged area of late endosomes/MVBs observed upon ORF3a expression (Fig. 1E–G, Supplementary Figs. S1K, S1L). In line with the reduced recycling of CI-M6PR from endosomes to the TGN, the transport of the newly synthesized CI-M6PR cargo, cathepsin Z (lysosomal hydrolase), from the Golgi to lysosomes was significantly impaired in ORF3a-expressing cells (Fig. 7E, F). These findings suggest that in addition to impairing the fusion of endosomes and autophagosomes, lysosomes also contain fewer hydrolases and are therefore less degradative upon ORF3a expression. Interestingly, several unrelated pathogenic effectors of bacterial and viral pathogens target the retromer complex and its interaction partner, Rab7 GAP TBC1D5, which in turn alters the Rab7 GTPase cycle and cargo recycling from maturing endosomes[50–52].

How does ORF3a expression promote Rab7 activation? The Rab GTPase cycle is regulated by guanine nucleotide exchange factors (GEFs), which catalyze the nucleotide exchange of bound GDP with 10-

fold more abundant GTP, and GTPase activating proteins (GAPs), which accelerate the hydrolysis of GTP γ-phosphate to generate GDP[53]. We found that, along with Vps39, ORF3a forms a complex with Rab7 GAP and TBC1D5 and impairs the interaction between TBC1D5 and Rab7 (Fig. 5H–L, and Supplementary Fig. S4L). We predicted that reduced binding of Rab7 to TBC1D5 leads to stabilization of active Rab7 in ORF3a-expressing or SARS-CoV-2-infected cells and enhanced binding of Rab7 to its effectors PLEKHM1 and RILP (Fig. 2I, J, G, and see the model in Fig. 9). Interestingly, from the viewpoint of early to late endosome maturation and fusion with lysosomes, we detected an interaction between the Rab7 GAP TBC1D5 and the HOPS subunit Vps39 (Supplementary Fig. S4J). Vps39 also interacts with Rab2, which is implicated in both late endosome and autophagosome fusion with lysosomes[54]. We propose a model in which Rab7 is displaced by the interaction of Vps39 with TBC1D5, followed by interaction with Rab2, which promotes HOPS-dependent late endosomal tethering and fusion with lysosomes.

Finally, using an approach in which endocytosed paramagnetic particles were isolated at various times, and their maturation status was analyzed by probing for early and late endosomal markers, we found that the Rab7 to Arl8b transition was strongly abrogated in ORF3a-expressing cells (Fig. 8G). These results suggest that hyperactivated Rab7 blocks the acquisition of Arl8b on endosomes. In agreement with these results, we noted that Arl8b colocalization with Rab7 was reduced (Fig. 8A–D); interestingly, the motility of Arl8b was enhanced in ORF3a-expressing cells (Fig. 8H–K), while as expected, the motility of the Rab7 compartments was significantly reduced (Fig. 8H, I, L, M). A previous study showed that SARS-CoV-2 uses Arl8b-positive lysosomes for cellular exocytosis[4]. Moreover, ORF3a has been shown to promote BORC lysosomal localization, which subsequently mediates Arl8b recruitment to lysosomes for exocytosis. Interestingly, this role of ORF3a requires binding to Vps39[17]. Our findings suggest that ORF3a-mediated constitutive activation of Rab7 blocks the fusion of immobile perinuclear Rab7 endosomes with Arl8b-positive compartments, and likely promotes peripheral motility and secretion of Arl8b compartments. This, in turn, could facilitate viral egress from the Arl8b-positive compartments.

Taken together, the findings presented in this study show that ORF3a is a major virulence factor that modulates and disrupts the properties and function of host late endocytic compartments by altering the activation status of Rab7 (Fig. 9), a master regulator of late endosomes, and lysosomal positioning and function.

# Methods
## Cell culture
HeLa, HEK293T, and A549 (from ATCC), Vero E6 (from the NCCS Pune Cell Repository), and HEK293T-hACE2 (HEK293T cells expressing

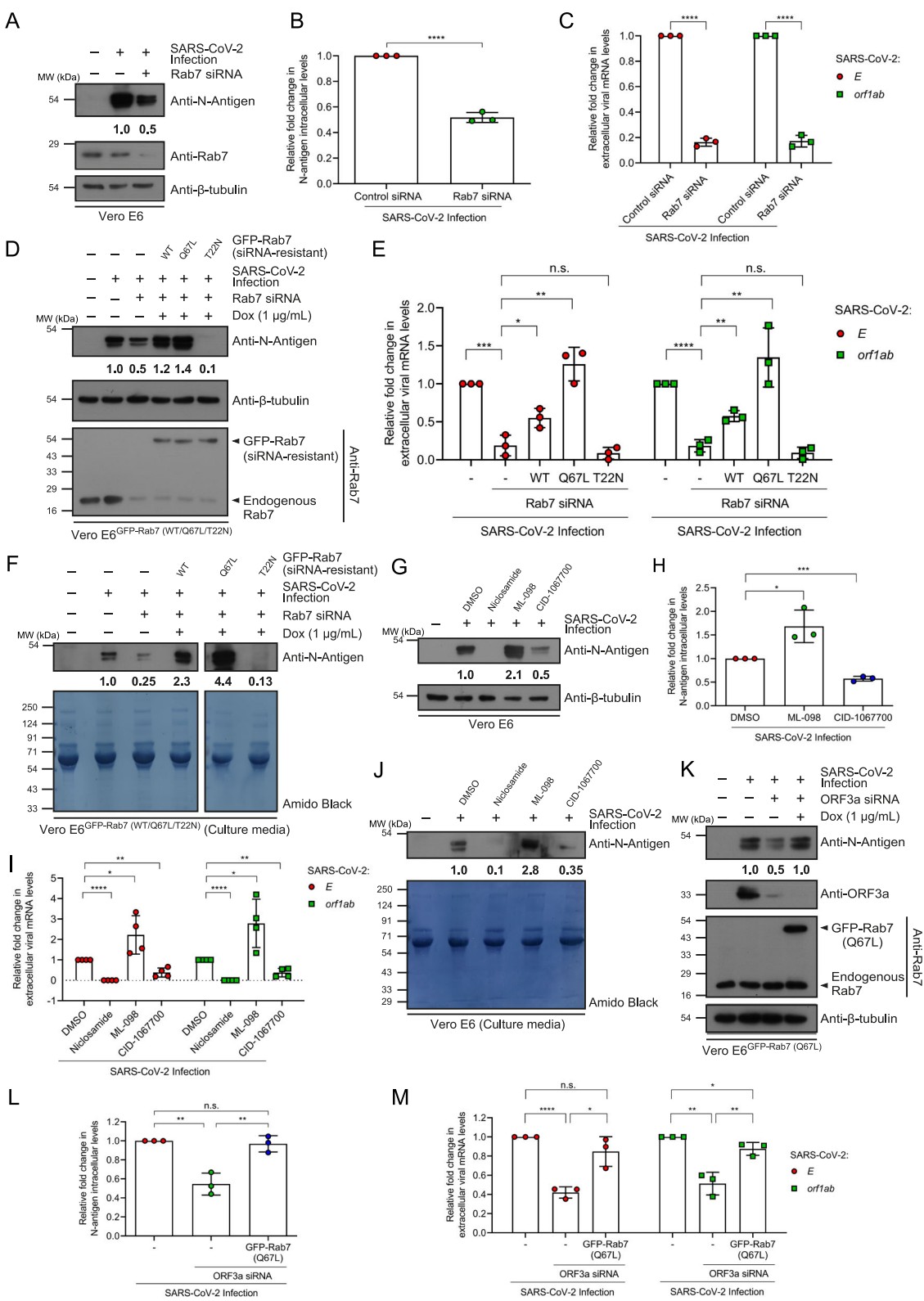

human angiotensin-converting enzyme 2 (ACE2); NR-52511; from BEI Resources) cells were cultured in DMEM (Gibco) supplemented with 10% FBS (Gibco) in a humidified cell culture incubator (Thermo Scientific) with 5% $CO_2$ at 37 °C.

HeLa cells stably expressing ORF3a-2xStrep under a doxycycline (Dox)-inducible system (HeLa$^{ORF3a-Strep}$) and A549 cells stably expressing myc-ACE2 (A549$^{myc-ACE2}$) were generated using a lentiviral transduction method, as described previously[49,55]. Briefly, lentiviruses were

produced by transfecting HEK293T cells with pLVX-TetOne-Puro-SARS-CoV-2-ORF3a-2xStrep and pCDH-CMV-EF1-Puro-myc-Human ACE2 lentiviral plasmids, along with VSV-G and dR8.91 packaging plasmids, respectively, to generate HeLa$^{ORF3a-Strep}$ and A549$^{myc-ACE2}$ cell lines. At 24 h post transfection, the medium was replaced with DMEM supplemented with 30% FBS. After 24 h, the cell culture supernatant containing the virus particles was collected in a conical tube, and fresh DMEM containing 30% FBS was added. After 24 h, the supernatant

**Fig. 6 | The GTP-bound form of Rab7 promotes SARS-CoV-2 infection. A** Lysates of SARS-CoV-2 infected and control or Rab7 siRNA transfected cells were IB for the indicated proteins, $n = 3$. The values in (**A**) and graph (**B**) represent the densitometric analysis of N-antigen levels normalized to β-tubulin, $n = 3$, $p < 0.0001$. **C** Relative expression of the $E$ and $orf1ab$ genes measured in the culture supernatants, $n = 3$, $p < 0.0001$($E$ and $orf1ab$ genes). **D** Protein levels of N-antigen in the lysates of SARS-CoV-2-infected cells transfected with control or Rab7 siRNA and expressing indicated proteins. The values represent densitometric analysis of the levels of N-antigen normalized to β-tubulin, $n = 3$. **E** Relative expression of $E$ and $orf1ab$ genes measured in the media, $n = 3$. For control versus Rab7 siRNA, $p = 0.0005$($E$), $p < 0.0001$($orf1ab$); for Rab7 (WT) versus Rab7 siRNA, $p = 0.0278$($E$), $0.0034$($orf1ab$); for Rab7 (Q67L) versus Rab7 siRNA, $p = 0.0021$($E$), $0.0072$($orf1ab$); for Rab7 (T22N) versus Rab7 siRNA, $p = 0.0321$($E$), $0.2202$($orf1ab$). **F** Protein levels of N-antigen in the media. The values represent the densitometric analysis of the levels of extracellular N-antigen, $n = 1$. **G** Lysates of cells infected with SARS-CoV-2 and treated with indicated chemical compounds were IB for the indicated proteins, $n = 3$. The values in (**G**) and graph (**H**) represent densitometric analysis of N-antigen levels normalized to β-tubulin, $n = 3$, $p = 0.0264$(ML-098), $0.0001$(CID-1067700). **I** Relative expression of the $E$ and $orf1ab$ genes measured in the media, $n = 3$. For niclosamide, $p < 0.0001$($E$ and $orf1ab$ genes); for ML-098, $p = 0.0405$($E$), $0.0233$($orf1ab$); for CID-1067700, $p = 0.0015$($E$), $0.001$($orf1ab$). **J** Protein levels of N-antigen in the media. The values represent a densitometric analysis of the levels of extracellular N-antigen, $n = 1$. **K** Protein levels of the N-antigen in the lysates of SARS-CoV-2-infected cells transfected with control or ORF3a siRNA and expressing GFP-Rab7 (Q67L), $n = 3$. The values in (**K**) and graph (**L**) represent densitometric analysis of the levels of N-antigen normalized to β-tubulin, $n = 3$, $p < 0.0001$(control versus ORF3a siRNA), $0.0111$(GFP-Rab7 (Q67L) versus ORF3a siRNA), $0.164$(control versus GFP-Rab7 (Q67L)-ORF3a siRNA). **M** Relative expression of the $E$ and $orf1ab$ genes measured in the media, $n = 3$. For control versus ORF3a siRNA, $p < 0.0001$($E$), $p = 0.0021$($orf1ab$); for GFP-Rab7 (Q67L) versus ORF3a siRNA, $p = 0.0111$($E$), $p = 0.01$($orf1ab$); for control versus GFP-Rab7 (Q67L)-ORF3a siRNA, $p = 0.164$($E$), $p = 0.335$($orf1ab$).

containing the viral particles was collected and combined with the previously collected cell culture supernatant. The supernatant was centrifuged at $1000 \times g$ for 5 min to remove cell debris before being carefully transferred to a new conical tube. The lentiviral particles in the media were concentrated by adding a LentiX concentrator (Takara) according to the manufacturer's protocol and kept at 4 °C overnight. The mixture was centrifuged at $1500 \times g$ for 45 min, and the supernatant was discarded. The pellet was gently dissolved in DMEM. To generate the cell lines, HeLa and A549 cells were cultured in a 35-mm dish (Corning) in DMEM with 10% FBS and transduced with lentivirus-containing medium along with 8 μg/mL polybrene (Sigma-Aldrich). After 24 h, the medium was discarded and replacement medium containing 3 μg/mL puromycin was added for selection. The expression of the desired proteins in stable cell lines was evaluated using SDS-PAGE and immunoblotting. Unless otherwise specified, the expression of ORF3a in the HeLa$^{ORF3a-Strep}$ cell line was induced by the addition of 1 μg/mL Dox (Sigma-Aldrich) for 24 h.

Using the same strategy as described above, Vero E6 cells stably expressing siRNA-resistant ORF3a (WT), ORF3a (S171E), or ORF3a (W193R) with a C-terminal HA tag under a Dox-inducible system were generated, and 8 μg/mL puromycin was added for selection. In addition, a similar experimental procedure was followed to generate Vero E6 cells stably expressing N-terminal GFP-tagged siRNA-resistant Rab7 (WT, GTP-locked (Q67L), or GDP-locked (T22N)) under a Dox-inducible system.

Each cell line was cultured for no more than 15 passages and routinely tested for mycoplasma contamination using the MycoAlert Mycoplasma Detection Kit (Lonza).

## DNA constructs, antibodies, chemicals and siRNAs

All the DNA constructs and antibodies used in this study are listed in Supplementary Tables I and II, respectively. Most of the chemicals used in this study were purchased from Sigma-Aldrich. Alexa-fluor-conjugated-dextran, LysoTracker, LysoSensor dyes, EGF, and DAPI were purchased from Molecular Probes. The SiR-Lysosome Kit was purchased from Cytoskeleton, Inc., and the self-quenched BODIPY-FL conjugate of BSA was purchased from BioVision. Water-based ferro-fluid (EMG 508) was purchased from Ferrotec Corporation. The chemical compounds, ML-098 and CID-1067700, were purchased from Cayman Chemicals.

The siRNA oligos for gene silencing were purchased from Dharmacon (Horizon Discovery) and prepared according to the manufacturer's instructions. Transient transfection of siRNAs was performed using the DharmaFECT 1 reagent according to the manufacturer's instructions. The following siRNAs were used in this study: control siRNA, 5′-TGGTTTACATGTCGACTAA-3′; ORF3a siRNA, 5′-GAGAATCTTCACAATTGGAACTGTA-3′[56]; Rab7 siRNA, 5′-CTAGATAGCTG GAGAGATG-3′ and Vps39 siRNA 5′-CATTGCAGTGTTGCCTCGA-TATG-3′.

## Transfection, immunofluorescence, and live-cell imaging

Using X-tremeGENE HP DNA transfection reagent (Roche), cells grown on glass coverslips (VWR) were transfected with the desired plasmids for 16–18 h. Cells were fixed with 4% paraformaldehyde (PFA) in PHEM buffer (60 mM PIPES, 10 mM EGTA, 25 mM HEPES, 2 mM MgCl$_2$, and a final pH of 6.8) at room temperature (RT) for 10 min. Following fixation, the cells were incubated with a blocking solution (0.2% saponin + 5% NGS in PHEM buffer) for 30 min at RT, followed by three 5-min washes in 1X PBS. After blocking, the cells were incubated with primary antibodies in staining solution (PHEM buffer + 0.2% saponin + 1% NGS) for 1 h at RT, washed three times with 1X PBS for 5 min each, and then incubated for 30 min with Alexa fluorophore-conjugated secondary antibodies in staining solution. Fluoromount G (Southern Biotech) was used to mount the coverslips, and confocal images were acquired using a Carl Zeiss 710 confocal laser scanning microscope or LSM 980 Elyra 7 super-resolution microscope equipped with a plan apochromat 63/1.4 NA oil immersion objective. For image acquisition, the ZEN 2012 v. 8.0.1.273 (ZEISS) software was used. All micrographs were captured such that there was minimal or no pixel saturation. Representative confocal images were adjusted for brightness and contrast using Fiji software or Adobe Photoshop CS.

To perform immunofluorescence staining of ORF3a in Vero E6 cells infected with SARS-CoV-2 (Supplementary Fig. S1H), the cells were permeabilized for 5 min on ice with 0.05% saponin in PHEM buffer before the fixation step. To perform cathepsin D immunofluorescence staining in HeLa cells (Supplementary Fig. S1C), a methanol-based fixation method was used. Briefly, cells were fixed with ice-cold methanol for 10 min by incubating the coverslips at −20 °C. After fixation, the cells were washed once with 1X PBS, and the saponin-based staining protocol described previously was used.

For live-cell imaging experiments, cells were seeded on a glass-bottom tissue culture-treated cell imaging dish (ibidi). For vesicle tracking experiments, cells were transfected with the desired plasmids, and 24 h post-transfection, the medium was replaced with phenol red-free DMEM medium (Gibco) containing 10% FBS, and the dish was placed in a live-cell imaging chamber maintained at 37 °C with a 5% CO$_2$ supply. Time-lapse imaging was performed using a Carl Zeiss 710 confocal laser scanning microscope or LSM 980 Elyra 7 super-resolution microscope with a plan apochromat 63×/1.4 NA oil immersion objective. Live-cell imaging movies were acquired using Zen Black 2012 software (ZEISS), and final adjustments for brightness and contrast were performed using the ImageJ software.

To label lysosomes using Lysotracker dye, cells were incubated in phenol red-free complete DMEM (Gibco) containing LysoTracker Red

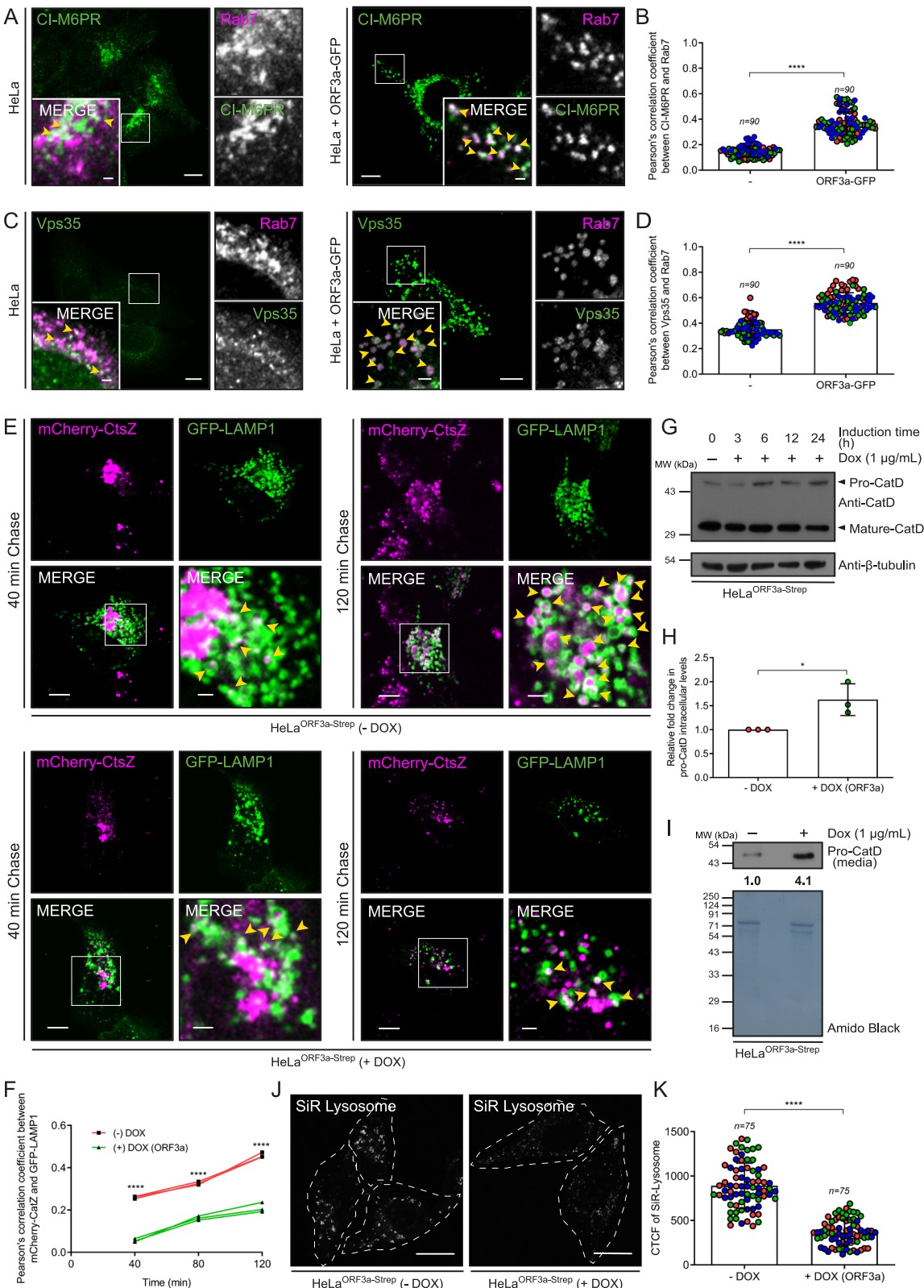

DND-99 (100 nM; Molecular Probes) for 1 h at 37 °C in a cell culture incubator. Cells were washed three times with 1X PBS to remove excess probe, followed by fixation with 4% PFA in PHEM buffer, and confocal imaging was performed as described above.

To analyze the activity of cathepsin D in lysosomes, the SiR-Lysosome probe was used, and uptake was performed according to the manufacturer's instructions. Briefly, the cells were incubated for 1 h at 37 °C in phenol red-free complete DMEM (Gibco) containing SiR-Lysosome (1 μM). To remove excess probe, cells were rinsed three times with 1X PBS, fixed with 4% PFA in PHEM buffer, and confocal imaging was performed as described above. Images were imported into the Fiji software, and the corrected total cell fluorescence (CTCF) of the SiR-Lysosome probe was calculated using the formula CTCF = integrated density − (area x background mean fluorescence).

**Fig. 7 | ORF3a blocks CI-M6PR recycling from late endosomes to the TGN and impairs lysosomal sorting of newly synthesized hydrolases. A** Representative confocal images showing HeLa (wild-type) and ORF3a-GFP-expressing cells immunostained for indicated proteins. **B** The colocalization of CI-M6PR with Rab7 was measured using PCC, $n = 90$ cells examined over three independent experiments, $p < 0.0001$. **C** Representative confocal images showing HeLa (wild-type) and ORF3a-GFP-expressing cells immunostained for indicated proteins.
**D** Colocalization of Vps35 and Rab7 was measured using PCC, $n = 90$ cells examined over three independent experiments, $p < 0.0001$. **E** The RUSH assay was performed in untreated and Dox-treated (12 h) HeLa^ORF3a-Strep cells expressing Str-KDEL-IRES-SBP-mCherry-CtsZ and LAMP1-GFP. Live-cell imaging was performed at various time points after the addition of biotin to follow the trafficking of the cargo, mCherry-CtsZ, to LAMP1-GFP-positive compartments. Representative confocal micrographs of cells taken 40- and 120-min post-biotin addition are shown. **F** The

PCC was quantified for the cargo, mCherry-CtsZ, with LAMP1-GFP at different time points post-biotin addition, $n = 3$, $p < 0.0001$. **G** Lysates from untreated and Dox-treated HeLa^ORF3a-Strep cells for the indicated periods were IB with anti-Cathepsin-D (CatD) antibodies, $n = 3$. **H** Densitometric analysis of the pro-CatD normalized to those of β-tubulin, $n = 3$. **I** Immunoblotting of pro-CatD in the cell culture supernatant media of untreated and Dox-treated HeLa^ORF3a-Strep cells, $n = 3$.
**J** Representative confocal images of untreated and Dox-treated HeLa^ORF3a-Strep cells labeled with the SiR Lysosome probe for the detection of CatD activity, $n = 3$.
**K** Quantification of the average fluorescence intensity of the SiR Lysosome probe signal in untreated and Dox-treated HeLa^ORF3a-Strep cells, $n = 75$ cells examined over three independent experiments, $p < 0.0001$. Quantified results are presented as mean ± S.D. using unpaired two-tailed Student's $t$ test. Scale bars: 10 μm (main); 2 μm (inset).

## Image analysis and quantification

**Colocalization analysis.** For all colocalization analyses, Pearson's correlation coefficient (PCC) was determined using the JACoP module of Fiji software.

**Distribution of lysosomes.** Lysosome distribution was analyzed by measuring the fractional distance of lysosomes from the center of the cell using the "Plot Profile" feature of Fiji software, as previously described[22,57]. A line was traced from the center of the nucleus to the cell's periphery on a confocal micrograph. Next, all lysosomal (LAMP1-positive) marker fluorescence intensities and their corresponding distance values along the line were extracted using the plot profile tool. After calculating the signal threshold, the background pixels and their respective distances were excluded from the analysis. All remaining distances (corresponding to lysosomal pixels only) were converted to fractional distances by dividing all values by the total line distance.

**Analysis of the area and number of vesicles.** To measure the area and number of vesicles (PLEKHM1-positive or Rab7-positive) from confocal images, each micrograph was converted to an 8-bit maximum intensity projection using the Fiji program. The area and number of vesicles were measured using the "Analyze Particle" Tool. To measure the diameter of LAMP1-positive vesicles using confocal micrographs or the size of multi-vesicular and multi-lamellar compartments using TEM micrographs, the diameter of each individual compartment was measured manually in Fiji software using the "Line" tool to draw a straight line across the lysosome.

## Single-particle tracking

To perform single-particle tracking analysis of Rab7-positive or Arl8b-positive endosomes, HeLa^ORF3a-Strep cells seeded on glass-bottom tissue culture-treated live-cell imaging dishes (ibidi) were either left untreated or Dox-treated (1 μg/mL) for 8 h in complete DMEM medium in a cell culture incubator. After 8 h, the cells were co-transfected with GFP-Rab7 and Arl8b-tomato expressing plasmids and incubated further for 16 h in complete DMEM with or without Dox (1 μg/mL). Before the start of the time-lapse confocal imaging, cells were washed with 1X PBS, phenol red-free DMEM media (Gibco) supplemented with 10% FBS was added, and live-cell imaging was performed using a ZEISS LSM 980 Elyra 7 super-resolution microscope with a 63×/1.4 NA oil immersion objective. To measure the mobile fraction and average speed of Rab7- or Arl8b-positive endosomes from time-lapse images, the "TrackMate" plugin of Fiji software was used with the following parameters:

- Vesicle diameter, 1 μm
- Detector, DoG
- Initial thresholding, none
- Tracker, Simple LAP tracker
- Linking max distance, 2 μm

- Gap-closing max distance, 2 μm
- Gap-closing max frame gap, 2
- Filters, none

After imaging, all the data were exported to a Microsoft Excel spreadsheet (2016) for further analysis.

## Fluorescence recovery after photobleaching (FRAP)

To perform FRAP, HeLa^ORF3a-Strep cells seeded on glass-bottom tissue culture-treated live-cell imaging dishes (ibidi) were either left untreated or Dox-treated (1 μg/mL) for 8 h in complete DMEM medium in a cell culture incubator. After 8 h, the cells were transfected with a GFP-expressing Rab7 plasmid and incubated further for 16 h in complete DMEM with or without Dox (1 μg/mL). Before the start of the experiment, the cells were washed with 1X PBS, and phenol red-free DMEM (Gibco) supplemented with 10% FBS was added. FRAP analysis was conducted at 37 °C and 5% $CO_2$ in a live-cell imaging chamber using the FRAP module in a ZEISS LSM 980 Elyra 7 super-resolution microscope with a 63× oil-immersion objective and an NA 1.4. Three ROIs were marked in the cell using the software ROI selection tool. A single GFP-Rab7 endosome was selected for bleaching (ROI 1), the region outside the cell was selected as the background (ROI 2), and the region encompassing the entire cell (ROI 3). The bleaching parameters involved five pre-bleach scans of selected endosomes followed by photobleaching using a 488 nm laser line at 100% power with 150 iterations. The fluorescence intensities in the bleached area, background, and entire cell were scanned over time using a conventional laser power (lower than that used for bleaching). To analyze the recovery rate of the GFP-Rab7 endosome, the fluorescence intensities of the endosomes were corrected for background fluorescence and differences in protein expression levels between cells, as previously described[58]. The equation for the normalization of fluorescence intensities is as follows:

$$F(t)_{norm} = \frac{F(t)_{ROI} - F_{bkgd}}{F(t)_{cell} - F_{bkgd}} \times \frac{F(i)_{cell} - F_{bkgd}}{F(i)_{ROI} - F_{bkgd}}$$

$F(t)_{ROI}$ is the fluorescence intensity of bleaching ROI intensity at any given time point.
$F_{bkgd}$ is the fluorescence intensity of background.
$F(i)_{cell}$ is the initial fluorescence intensity of the cell (at time=0),
$F(i)_{ROI}$ is the initial fluorescence intensity of the bleaching ROI.
FRAP recovery curve was plotted with normalized fluorescence intensities $F(t)_{norm}$ against time. Mobile fraction ($M_f$) and half time of recovery $t_{1/2}$ were calculated for each FRAP curve and averaged to get final values.

Mobile fraction was calculated as:

$$M_f = \frac{(F_\infty - F_0)}{(F_i - F_0)} \times 100$$

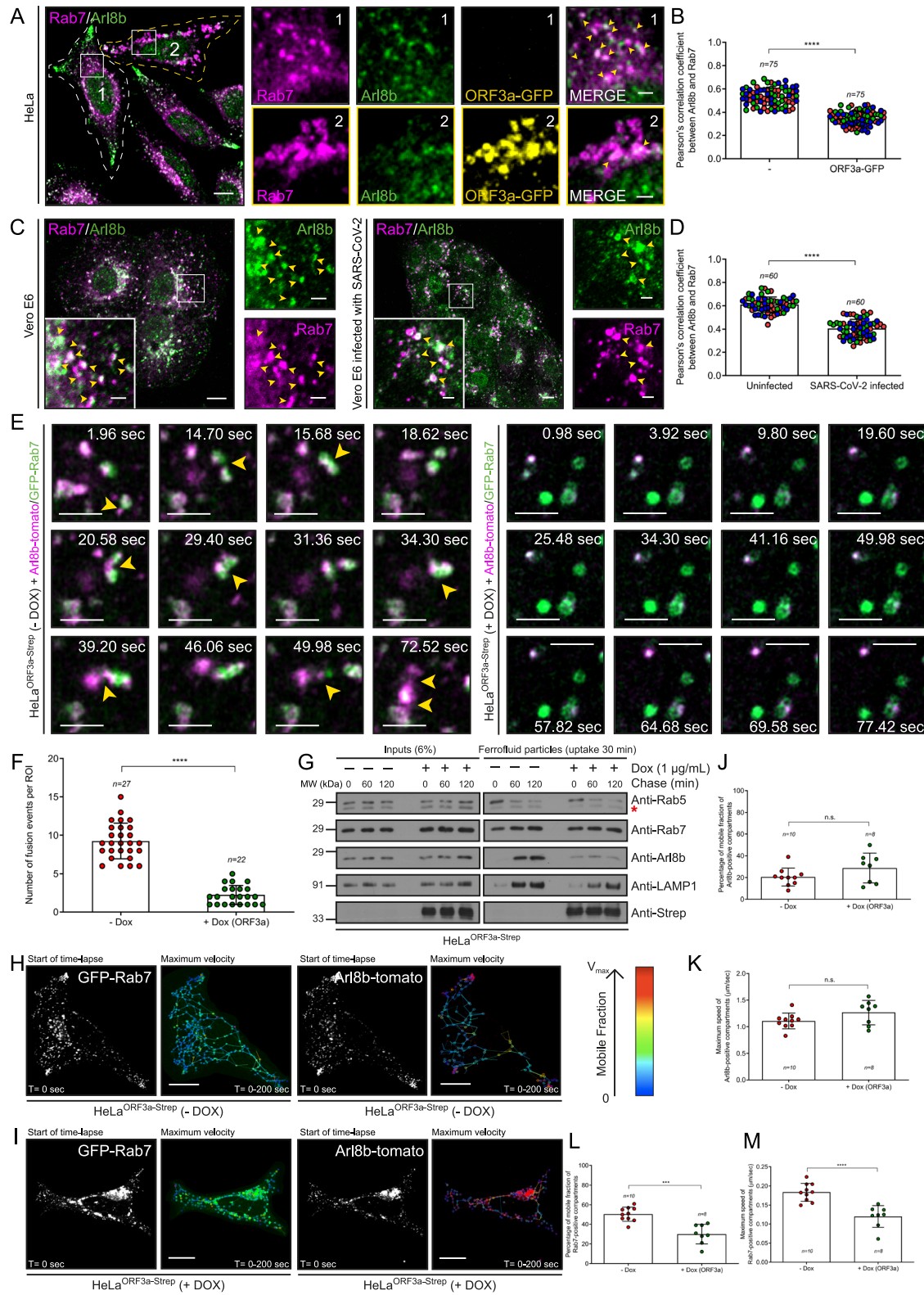

Half time of recovery was calculated as:

$$F(t) = [F_0 + F_\infty(t/t_{1/2})]/[1 + (t/t_{1/2})]$$

## Cell lysates, co-immunoprecipitation, and immunoblotting

For lysate preparation, cells were lysed in ice-cold RIPA lysis buffer 10 mM Tris-Cl pH 8.0, 1 mM EDTA, 0.5 mM EGTA, 1% Triton X-100,

0.1% SDS, 0.1% sodium deoxycholate, 140 mM NaCl supplemented with PhosSTOP (Roche) and a protease inhibitor cocktail (Sigma-Aldrich). The samples were incubated on ice for 2 min, followed by a 30-s vortex, and this cycle was repeated at least five times prior to centrifugation at $16,000 \times g$ for 10 min at 4 °C. Using a bicinchoninic acid assay kit (Sigma-Aldrich), the amount of protein was quantified from the clear supernatants collected in new microcentrifuge tubes. Protein samples were prepared for immunoblotting analysis by

**Fig. 8 | ORF3a blocks the fusion of Rab7-positive and Arl8b-positive late endosomes and lysosomes. A** Representative confocal images of HeLa (wild-type) and ORF3a-GFP-expressing cells immunostained for indicated proteins. The insets 1 and 2 mark untransfected and ORF3a-GFP-transfected cells, respectively. **B** The colocalization of Rab7 and Arl8b was evaluated using PCC, $n = 75$ cells examined over three independent experiments, $p < 0.0001$. **C** Representative confocal images of uninfected and SARS-CoV-2-infected Vero E6 cells immunostained for Arl8b and Rab7. **D** The colocalization of Rab7 with Arl8b was measured using PCC, $n = 60$ cells examined over three independent experiments, $p < 0.0001$. **E** Representative micrographs from live-cell imaging experiments performed on untreated and Dox-treated HeLa$^{ORF3a\text{-}Strep}$ cells expressing GFP-Rab7 and Arl8b-tomato. The arrows highlight the fusion between GFP-Rab7 and Arl8b-tomato vesicles; see Supplementary Movie 2. **F** Quantification of the number of fusion events per ROI (region of interest), $n = 27$ (-DOX) and $n = 22$ (+ DOX), $p < 0.0001$. **G** Ferrofluid (FF) uptake was performed in untreated and Dox-treated HeLa$^{ORF3a\text{-}Strep}$ cells for 20 min at 37 °C

(pulse), followed by various periods of time (chase). At the indicated time points, the cells were homogenized, and the FF-containing cellular compartments were purified and immunoblotted for the presence of the indicated proteins, $n = 1$. The asterisk (*) denotes a non-specific signal detected using the anti-Rab5 antibody. Representative live-cell imaging micrographs of untreated (**H**) and Dox-treated (**I**) HeLa$^{ORF3a\text{-}Strep}$ cells expressing GFP-Rab7 and Arl8b-tomato were captured at the start of time-lapse imaging ($T = 0$ s). Single-particle tracking analysis of GFP-Rab7 and Arl8b-tomato was performed until $T = 200$ s with color coding to show the maximum velocity ($V_{max}$; blue, immobile; red, maximum mobility), $n = 10$ (-DOX) and $n = 8$ (+DOX) cells. **J–M** Graphs (**J**) and (**L**) represent the mobile fractions of Arl8b-tomato and GFP-Rab7 vesicles, respectively. Graphs (**K**) and (**M**) represent the maximum speeds of the Arl8b-tomato and GFP-Rab7 vesicles, respectively, $n = 10$ (-DOX) and $n = 8$ (+DOX) cells, $p = 0.1343$ (**J**), 0.0952 (**K**), 0.0001 (**L**), < 0.0001 (**M**). Quantified results are presented as mean ± S.D. using unpaired two-tailed Student's $t$ test. Scale bars: 10 μm (main); 2 μm (inset).

boiling them in a 4X sample loading buffer and then loaded onto SDS-PAGE.

To perform the co-immunoprecipitation assay, previously described methodology was used[59]. Briefly, cells were lysed in ice-cold TAP lysis buffer (20 mM Tris-Cl pH 8.0, 150 mM NaCl, 0.5% NP-40, 1 mM MgCl$_2$, 1 mM Na$_3$VO$_4$, 1 mM NaF, 1 mM PMSF, and protease inhibitor cocktail) on rotation (Hula Mixer, Thermo Scientific) for 30 min at 4 °C. The cell lysate was centrifuged at 13,000 rpm for 10 min at 4 °C and the post-nuclear supernatant (PNS) was collected. PNS was incubated for 3 h at 4 °C with the indicated antibody-conjugated agarose beads, followed by four washes with TAP wash buffer (20 mM Tris-Cl pH 8.0, 150 mM NaCl, 0.1% NP-40, 1 mM MgCl$_2$, 1 mM Na$_3$VO$_4$, 1 mM NaF, and 1 mM PMSF). For the co-immunoprecipitation assay involving TBC proteins, a TAP lysis buffer containing 5 mM MgCl$_2$ was used. Protein complexes were eluted by boiling the beads in a 2X sample loading buffer at 100 °C for 10 min. The samples were then subjected to SDS-PAGE for western blotting.

For immunoblotting, protein samples separated by SDS-PAGE were transferred onto polyvinylidene fluoride (PVDF) membranes (Bio-Rad), followed by overnight incubation at 4 °C in blocking buffer (10% skim milk (BD Difco) prepared in 1X PBS containing 0.05% Tween 20 (Sigma-Aldrich)). After washing with 0.05% PBS-Tween 20, the blot was incubated with a primary antibody solution prepared in 0.05% PBS-Tween 20 for 2 h at RT. The membranes were washed three times for 10 min each with 0.05% PBS-Tween 20 and further incubated with an HRP-conjugated secondary antibody solution prepared in 0.05% PBS-Tween 20 for 1 h at RT. After the secondary antibody step, the membranes were washed twice for 10 min with 0.3% PBS-Tween 20 and once with 0.05% PBS-Tween 20. The blots were developed by a chemiluminescence-based method (ECL Plus Western Blotting Substrate; Thermo Scientific) using X-ray films (Carestream). ImageJ software was used to perform densitometry analysis of the immunoblots.

For estimation of released cathepsin D in the cell culture supernatant, HeLa$^{ORF3a\text{-}Strep}$ cells were seeded in culture dishes, and ORF3a expression was induced using 1 μg/ml Dox for 8 h in complete DMEM. Untreated (control) and Dox-treated cells were incubated in Opti-MEM (reduced serum medium; Gibco) for 16 h. The media supernatant was collected and centrifuged at $582 \times g$ at 4 °C for 2 min to remove cellular debris, transferred to a fresh tube, and four volumes of acetone were added to it, mixed gently, and kept at −20 °C for 2 h to precipitate all proteins. The medium was again centrifuged at $30,000 \times g$ for 15 min to separate the pellets containing proteins. The pellet was gently dissolved in 1X PBS, and samples were prepared by boiling them in 2X sample loading buffer for immunoblotting analysis, as described previously.

## Recombinant protein purification and GST-pulldown assay

In this study, all GST-tagged recombinant proteins were expressed and purified from *E. coli* Rosetta (DE3) strain. For the setting up of primary

cultures, a single transformed colony was inoculated into Luria–Bertani (LB) broth containing the appropriate antibiotics (ampicillin and chloramphenicol) and incubated at 37 °C in a shaking incubator. After 8–12 h of incubation, 1% of primary cultures were used as inoculum to establish secondary cultures, which were then incubated at 37 °C with shaking until the absorbance at 600 nm reached 0.4–0.6. To induce protein expression, 0.3 mM IPTG (Sigma-Aldrich) was added to the cultures, followed by 16 h of shaking incubation at 16 °C. Bacterial cultures were centrifuged at $3542 \times g$ for 10 min, rinsed once with 1X PBS, and resuspended in bacterial cell lysis buffer (20 mM Tris-Cl pH 8.0, 150 mM NaCl) containing a protease inhibitor tablet (Roche) and 1 mM PMSF (Sigma-Aldrich). Sonication of the bacterial cell suspension was followed by 45 min of centrifugation at $15,557 \times g$ and 4 °C. The clear supernatants were incubated with glutathione resin (Gbiosciences) for 2–3 h at 4 °C to allow binding of GST-tagged proteins. To remove impurities, the beads were rinsed a minimum of six times with a wash buffer (20 mM Tris-Cl pH 8.0, 300 mM NaCl).

For the GST-pulldown assay using mammalian cells as a source of lysates, the cells were lysed in ice-cold TAP lysis buffer (20 mM Tris-Cl pH 8.0, 150 mM NaCl, 0.5% NP-40, 1 mM MgCl$_2$, 1 mM Na$_3$VO$_4$, 1 mM NaF, 1 mM PMSF, and protease inhibitor cocktail) at 4 °C for 10 min, followed by centrifugation at 16,627×g for 10 min at 4 °C. Lysates were collected and incubated with GST protein or GST-fusion protein bound to the glutathione resin for 3–4 h at 4 °C with Hula mixer rotation. After incubation, beads were rinsed a minimum of six times with TAP lysis buffer, and protein complexes were eluted by boiling samples in 2X sample loading buffer at 100 °C for 10 min before loading them onto SDS-PAGE gel and immunoblotting, as previously described.

## SARS-CoV-2 virus preparation and infection

All experiments involving the handling of SARS-CoV-2 were performed in the BSL3 facility according to the institutional biosafety guidelines and institutional ethics guidelines (CSIR/IMTECH/IBSC/2020/J17 and CSIR/IMTECH/IBSC/2020/J23). The SARS-CoV-2 strain used in this study was isolated from a nasal swab sample of a COVID-19-confirmed patient and cultured using the Vero E6 cell line, according to established methods[60]. SARS-CoV-2 was confirmed by whole-genome sequencing and the sequence was submitted to GenBank (accession number: EPI_ISL_11450498). An aliquot of the virus from passage 1 was used to inoculate a 25-mm cell culture flask containing Vero E6 cells at 80-90% confluent in 5 mL of medium. After extensive cytopathic effects, the virus suspension was harvested, clarified, and stored at −80 °C. The culture supernatant was analyzed by quantitative real-time PCR to check for virus growth. The virus stock was further titrated using Vero E6 cells for plaque-forming units (pfu) per mL and was estimated to be $5 \times 10^7$ pfu/mL. The virus stock was stored at −80 °C until further use, and the same stock was used for all experiments.

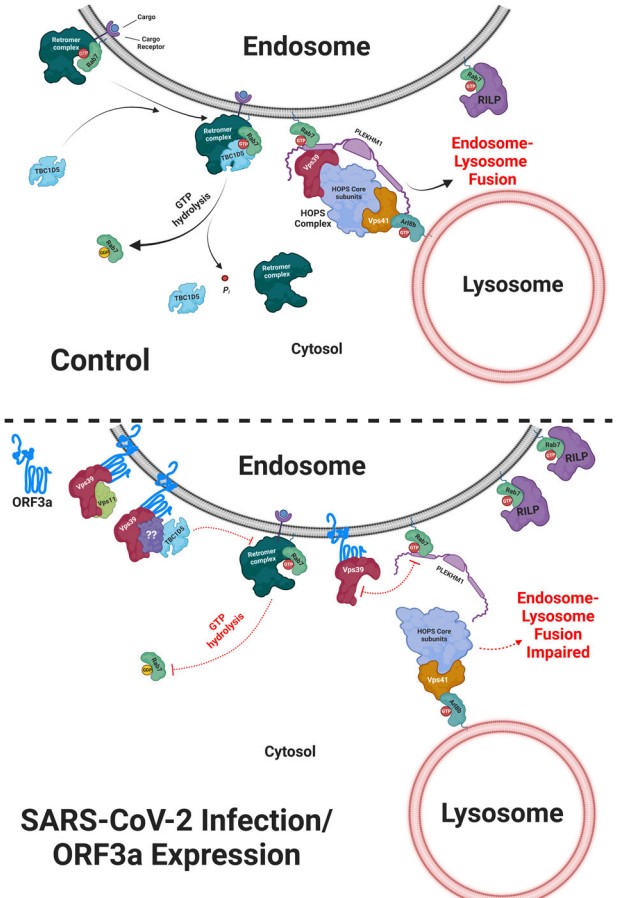

**Fig. 9 | Schematic illustrating the role of the interaction between SARS-CoV-2 ORF3a and Vps39 in blocking Rab7 GTP-GDP cycling.** The late endosomal G protein Rab7 interacts with its effectors RILP and PLEKHM1 to mediate late endosomal positioning and fusion with lysosomes, respectively. PLEKHM1 is a dual-binding partner of Rab7 and Arl8b that binds to the Vps39 subunit of the tethering factor HOPS complex, while the Vps41 subunit directly binds to Arl8b on lysosomes. Recruitment of the Rab7 GAP, TBC1D5, by the trimeric retromer complex to late endosomes leads to GTP hydrolysis of Rab7, which leads to the dissociation of the Rab7 effectors RILP and PLEKHM1 from late endosomal membranes. Upon SARS-CoV-2 infection, the viral protein ORF3a localizes to late endosomes and lysosomes and directly interacts with Vps39. ORF3a abrogates the interaction of Vps39 with PLEKHM1 and promotes its interaction with TBC1D5. ORF3a sequesters TBC1D5 and impairs its binding to Rab7, leading to increased levels of GTP-bound Rab7. By inhibiting Vps39 and PLEKHM1 binding, ORF3a likely blocks the fusion of Rab7-positive and Arl8b-positive compartments and abrogates endocytic and autophagic cargo degradation. The model shown in the figure was created using BioRender.com.

For GST-pulldown assays and immunofluorescence experiments performed on SARS-CoV-2 infection, Vero E6 or A549$^{myc-ACE2}$ cells seeded in 35-mm cell culture dishes or glass coverslips were infected with SARS-CoV-2 at an M.O.I. of 1:100 in DMEM media without FBS for 1 h at 37 °C and 5% $CO_2$ in a cell culture incubator. Post-infection, cells were washed twice with 1X PBS, and fresh complete DMEM supplemented with 10% FBS was added to the cultures and incubated for 48 h or 72 h at 37 °C and 5% $CO_2$ in a cell culture incubator. The cells were either harvested or fixed with 4% PFA in PHEM buffer for 10 min for GST-pulldown assays or immunofluorescence staining, respectively, as previously described.

To determine the efficacy of ML-098 and CID-1067700 as Rab7 activators and inhibitors, respectively, Vero E6 cells were treated with the chemical compounds at a concentration of 100 μM for 48 h before GTP-Rab7 levels was measured by GST-pulldown using the GST-RILP-Rab7 binding domain (mR7BD) as previously described.

For co-immunoprecipitation experiments, HEK-293T-hACE2 cells were infected with SARS-CoV-2 virus at an M.O.I. of 1:500 in DMEM without FBS for 1 h at 37 °C and 5% $CO_2$ in a cell culture incubator. Post-infection, cells were washed twice with 1X PBS, and fresh complete DMEM supplemented with 10% FBS was added to the cultures and incubated for 24 h at 37 °C and 5% $CO_2$ in a cell culture incubator. After incubation, the cells were processed for co-immunoprecipitation assays as previously described.

For all siRNA-related experiments, Vero E6 cells were infected with SARS-CoV-2 at an M.O.I. of 1:500 in DMEM without FBS for 1 h at 37 °C and 5% $CO_2$ in a cell culture incubator. After infection, the cells were washed twice with 1X PBS, fresh complete DMEM supplemented with 10% FBS was added to the cultures, and the cells were incubated for 3 h at 37 °C and 5% $CO_2$ in a cell culture incubator. After incubation, the cells were transfected with the desired siRNA oligonucleotide and further incubated for 48 h in complete DMEM supplemented with 10% FBS at 37 °C and 5% $CO_2$ in a cell culture incubator. In the case of experiments involving Dox treatment for inducing the expression of a desired gene, Dox (1 μg/mL) was added after 30 min of siRNA transfection and was maintained until the end of 48 h.

### Measurement of SARS-CoV-2 gene expression via qRT-PCR analysis and immunoblotting

To detect SARS-CoV-2 gene expression, Vero E6 cells were infected for 48 h, using the methodology described above. After infection, the cell culture medium was collected, and viral RNA was isolated using the GSure Viral RNA Isolation Kit (GCC Biotech), according to the manufacturer's instructions. SARS-CoV-2 *E* and *orf1ab* gene expression was detected and quantified using a DiAGSure (GCC Biotech) COVID-19 detection kit in a CFX96 Real-time system (Bio-Rad).

To measure the extracellular protein levels of the SARS-CoV-2 N-antigen, post-infection Vero E6 cells were cultured in complete DMEM supplemented with 0.1% FBS at 37 °C and 5% $CO_2$ in a cell culture incubator for 48 h. The media supernatant was collected and centrifuged at 582×g at 4 °C for 2 min to remove cellular debris, transferred to a fresh tube, and four volumes of acetone were added to it, mixed gently, and kept at −20 °C for 2 h to precipitate all proteins. The medium was again centrifuged at $30,000 \times g$ for 15 min to separate pellets containing proteins. The pellet was gently dissolved in 1X PBS, and samples were prepared by boiling them in 2X sample loading buffer for N-antigen immunoblotting analysis, as described previously.

For experiments involving the use of niclosamide as a positive control for the inhibition of SARS-CoV-2 replication (Fig. 6G–J), Vero E6 cells were pre-treated with niclosamide (10 μM) for 1 h, followed by SARS-CoV-2 infection. After infection, niclosamide was added to the culture medium at a concentration of 1 μM until the end of the experiment.

### Immuno-purification of lysosomes

As described previously[22,23], the Lyso-IP method was modified to immunopurify the lysosomes. HEK293T cells stably expressing TMEM192-2xFLAG were transfected for 16–18 h with the desired plasmids. Following transfection, cells were collected, rinsed with 1X PBS, and homogenized with ice-cold KPBS buffer (136 mM KCl, 10 mM $KH_2PO_4$, pH adjusted to 7.25 with KOH) using a Dounce homogenizer (20 strokes). Homogenized cells were collected in a new microcentrifuge tube and centrifuged at $1000 \times g$ for 2 min. The supernatant was incubated for 15 min at 4 °C with the indicated antibody-conjugated agarose beads using a Hula mixer. The beads were gently washed three times with KPBS buffer, and bound lysosomes were eluted in Laemmli buffer before SDS-PAGE and immunoblot analysis.

### Subcellular fractionation and lysosome enrichment

Using the Lysosome Enrichment Kit (Thermo Scientific), subcellular fractionation was performed. Briefly, the cell pellet was rinsed in PBS

and homogenized with 20 strokes of Dounce homogenizer on ice. The homogenate was centrifuged at $500 \times g$ for 10 min at 4 °C, and the post-nuclear supernatant was diluted with OptiPrep gradient medium (Sigma-Aldrich) to a final concentration of 15% OptiPrep. The sample was carefully layered on a discontinuous density gradient (17%, 20%, 23%, 27%, and 30%). The gradient was ultracentrifuged at $1,45,000 \times g$ for 4 h at 4 °C in a SW60 Ti swinging bucket rotor (Beckman Colter). After centrifugation, nine 400 μL fractions were collected from the top down. The collected fractions were centrifuged at $18,000 \times g$ for 20 min in an SW41 Ti rotor at 4 °C, and the obtained pellet was suspended in Laemmli buffer, boiled for 10 min, and analyzed using SDS-PAGE and immunoblotting.

## Membrane-Cytosol fractionation

Membrane-cytosol fractionation using cultured cells was performed as previously described[25]. Briefly, HeLa[ORF3a-Strep] cells, untreated or Dox-treated (1 μg/mL) for 24 h in complete DMEM, were harvested and homogenized in ice-cold homogenization buffer (25 mM HEPES, 100 mM NaCl, 1 mM EDTA pH 7.4, 1 mM PMSF, and protease inhibitor cocktail) by 30 vertical strokes using a glass dounce homogenizer (Sigma-Aldrich). The homogenate was centrifuged at $800 \times g$ for 10 min to remove non-lysed cells and debris. The supernatant was then ultracentrifuged at $10,8000 \times g$ (CS150NX, HIMAC Eppendorf) for 1 h at 4 °C, and the resultant supernatant was collected in a fresh microcentrifuge as the cytosolic fraction, whereas the pellet (membrane fraction) was dissolved in urea buffer (70 mM Tris-Cl pH 6.8, 8 M urea, 10 mM n-ethylmaleimide, 10 mM iodoacetamide, 2.5% SDS, and 0.1 M DTT) by incubating at 37 °C for 15 min. The membrane and cytosolic fractions were resuspended in 4X sample loading buffer by heating at 100 °C for 10 min, followed by SDS-PAGE and immunoblotting with the indicated antibodies, as described above.

## Purification of ferrofluid (FF)-containing endocytic compartments

Uptake and purification of the FF-containing endocytic compartments were performed as described previously, with modifications[48]. For a single 60-mm tissue culture dish, the FF-containing medium was prepared by adding 6 μL FF (EMG 508) to 1 mL of DMEM. The mixture was sonicated for 30 s, filter-sterilized using a 0.2 μm pore-size filter, and kept at 37 °C. For the uptake (pulse) of FF, HeLa[ORF3a-Strep] cells, untreated or Dox-treated (1 μg/mL for 24 h) seeded in 60-cm tissue culture dishes were washed once with 1X PBS and incubated in 1 mL of FF-containing DMEM for 20 min at 37 °C. Cells were washed three times with 1X PBS, complete DMEM was added, and cells were incubated for the indicated time (chase) at 37 °C. At the end of the chase period, cells were harvested and homogenized (20 strokes) in homogenization buffer (250 mM sucrose, 20 mM HEPES, 0.5 mM EGTA pH 7.2, and protease inhibitor cocktail) using a Dounce homogenizer (Sigma-Aldrich) on ice. Homogenates were centrifuged at $800 \times g$ for 5 min at 4 °C to prepare a postnuclear supernatant (PNS). The PNS were collected and incubated on a DynaMag-2 magnet (Invitrogen) for 45 min at 4 °C. Supernatants were removed and FF-containing endosomes were washed gently once with homogenization buffer and sediment by centrifugation. The resulting FF-containing endosomes were suspended in 1X Laemmli buffer, boiled for 10 min at 100 °C, and analyzed by SDS-PAGE and immunoblotting, as described earlier.

## Measurement of lysosome pH

The Lysosensor Yellow/Blue DND-160 was used to measure the pH of lysosomes, as previously described[22,61]. Briefly, the cells in suspension were incubated for 3 min at 37 °C with 2 μM Lysosensor Yellow/Blue DND-160 (Molecular Probes) in phenol red-free DMEM (Gibco) containing 10% FBS. Following two washes with 1X PBS to remove excess dye, cells were incubated for 10 min at 37 °C in isotonic pH calibration buffers (143 mM KCl, 5 mM glucose, 1 mM MgCl$_2$, 1 mM CaCl$_2$, 20 mM MES, 10 M Nigericin, and 5 M Monensin) varying from 4 to 6. Following the distribution of 10,000 cells per well in a black 96-well plate (Thermo Scientific), fluorescence readings were recorded at 37 °C using a 96-well plate multi-mode fluorescence reader (Tecan Infinite M-PLEX). The samples were excited at 340 and 380 nm to detect emitted light at 440 and 540 nm, respectively. A pH calibration curve was generated by plotting the ratio of fluorescence intensity between 340 and 380 nm against the respective buffer pH values.

## Measurement of proteolytic activity of lysosomes by flow cytometry

To determine the proteolytic activity of lysosomes, the cells were incubated for 2 h at 37 °C in phenol red-free DMEM (Gibco) containing 20 μg/mL BODIPY-FL-BSA (BioVision). After incubation, the medium was discarded and the cells were trypsinized, rinsed, resuspended in ice-cold 1X PBS, and then analyzed by flow cytometry[22,25]. A BD FACS Aria Fusion Cytometer and BD FACS Diva software version 8.0.1 (BD Biosciences) were used to acquire the samples. Data analysis was performed using the BD FlowJo version 10.0.1.

## Autophagic flux assay

Autophagic flux was determined by checking for the rescue of LC3b-II and p62 degradation by treating cells growing in complete DMEM (fed state) or in EBSS (starvation state) with the V-ATPase inhibitor bafilomycin A1 (BafA1; 100 nM; Sigma-Aldrich) for 2 h. After treatment, the cells were processed for immunofluorescence as described above or for lysate preparation using ice-cold RIPA buffer supplemented with a protease inhibitor. Equal amounts of lysates were loaded onto SDS-PAGE, transferred to PVDF, and immunoblotted with the indicated markers.

## EGFR degradation assay

The degradation of EGFR in the lysosomes was monitored as previously described[24]. Briefly, the indicated cell types were serum-starved for 1 h in DMEM. Cells were pulsed with unlabeled EGF (20 ng/mL; Molecular Probes) in complete DMEM at 37 °C for the indicated time periods. At the end of each time point, cell lysates were prepared using ice-cold RIPA buffer supplemented with protease inhibitors, and equal amounts of lysates were subjected to SDS-PAGE for immunoblotting of EGFR. To detect EGFR levels by immunofluorescence, the indicated cell types were cultured on glass coverslips and pulsed with unlabeled EGF (20 ng/mL) in complete DMEM at 37 °C for 15 min and chased for the indicated time points in complete DMEM media. At the end of each period, the cells were fixed and immunostained for internalized EGFR. Images were imported into Fiji software, and the corrected total cell fluorescence (CTCF) of internalized EGFR was measured using the formula CTCF = integrated density − (area × mean fluorescence of background).

## Dextran uptake assay

Dextran loading and delivery to lysosomes were performed as previously described, with minor modifications[49,62]. Briefly, to pre-label lysosomes, HeLa[ORF3a-Strep] cells, untreated or Dox-treated (1 μg/mL for 24 h), seeded on glass coverslips were incubated in phenol red-free complete DMEM media containing Alexa-Fluor 488-conjugated-dextran (green; Molecular Probes) for 12 h at 37 °C. Cells were washed once with phenol red-free complete DMEM and further incubated in phenol red-free complete DMEM containing Alexa-Fluor 568-conjugated-dextran (red; Molecular Probes) for the indicated time periods at 37 °C and 5% CO$_2$ in a cell culture incubator. At the end of the incubation period, the cells were washed with 1X PBS, fixed, and mounted as described earlier. The coverslips were immediately imaged using a confocal microscope. The colocalization of Alexa-Fluor 568-conjugated-dextran (red) with Alexa Fluor 488-conjugated-

dextran (green) containing lysosomes was measured using the "JACoP" plugin of Fiji software.

## Retention Using Selective Hooks (RUSH) transport assay

The RUSH assay was performed as previously described[25,63]. Briefly, HeLa[ORF3a-Strep] cells seeded on glass-bottom tissue culture-treated live-cell imaging dishes (ibidi) were either left untreated or Dox-treated (1 μg/mL) for 8 h in complete DMEM in a cell culture incubator. After 8 h, cells were co-transfected with Str-KDEL-IRES-SBP-mCherry-CtsZ (Cathepsin Z reporter) and LAMP1-GFP expressing plasmids and incubated for 16 h in complete DMEM with or without Dox (1 μg/mL). Before the start of the experiment, the cells were washed with 1X PBS, and phenol red-free DMEM (Gibco) supplemented with 10% FBS was added. Using a ZEISS 710 confocal microscope with a plan apochromat 63×/1.4 NA oil immersion objective, single-plane confocal images were acquired before and after the addition of biotin (final concentration 40 μM; Sigma-Aldrich), which causes a synchronous release of the reporter from the hook, for the indicated time points. Pearson's correlation coefficient between mCherry-CtsZ and LAMP1-GFP was quantified using the "JACoP" plugin of the Fiji software.

## Transmission electron microscopy (TEM)

The Harvard Medical School EM Facility (Boston, United States) processed the samples and performed the TEM. Untreated or Dox-treated HeLa[ORF3a-Strep] cells seeded in 60-cm tissue culture dishes were rinsed once with 1X PBS and fixed in TEM fixative buffer (2.5% glutaraldehyde/2.5% paraformaldehyde prepared in 0.1 M sodium cacodylate buffer, pH 7.4) for 2 h at RT. After fixation, samples were rinsed once in 0.1 M sodium cacodylate (pH 7.4) buffer. The cells were post-fixed for 30 min in 1% osmium tetroxide/1.5% potassium ferrocyanide, washed with water three times, incubated in 1% aqueous uranyl acetate for 30 min, followed by two washes in water and dehydration in graded alcohol (5 min per alcohol: 50, 70, 95, 2 100%). The cells were removed from the dish in propylene oxide, pelleted at $1741 \times g$ for 3 min, and infiltrated overnight in a mixture of propylene oxide and TAAB Epon (Marivac Canada) at a ratio of 1:1. The samples were then embedded in TAAB Epon and polymerized for 48 h at 60 °C. Ultrathin sections were cut with a Reichert Ultracut-S microtome, placed on lead citrate-stained copper grids, and examined with a JEOL 1200EX transmission electron microscope equipped with an AMT 2k charge-coupled camera.

## Statistical analysis

All the data are presented as the mean ± S.D. unless otherwise specified. Statistical significance was determined using a two-tailed Student's $t$ test (GraphPad Prism 8.0) to calculate $p$ values. For figures, $****p < 0.0001$, $***p < 0.001$, $**p < 0.01$, $*p < 0.05$, or n.s., not significant ($p > 0.05$) is shown. The accompanying figure legends show the exact $p$ values.

## Reporting summary

Further information on research design is available in the Nature Portfolio Reporting Summary linked to this article.

## Data availability

All relevant information supporting the findings of this study is presented in the manuscript and supplementary materials. A source file comprising raw data and western blot images that have not been cropped is included in the manuscript. Source data are provided with this paper.

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

## Acknowledgements

K.W., S.P., P.C. and G.K. acknowledges fellowship support from the Council of Scientific & Industrial Research (CSIR). A.S. acknowledges fellowship support from the Science and Engineering Research Board (SERB). This work was supported by grants from the Science and Engineering Research Board [SERB; CRG/2022/003266 and CVD/2020/000733], Department of Biotechnology (DBT)/Wellcome Trust India Alliance Intermediate Fellowship [IA/I/14/2/501543], and from CSIR-IMTECH intramural funding [OLP-183] to A.T. This research was supported by the DBT/Wellcome Trust India Alliance Senior Fellowship [IA/S/19/1/504270], DBT JA-NWBA [BT/HRD/NWBA/39/01/2018-19], and

intramural funding from IISER Mohali to M.S. R.R. acknowledges funding from SERB [IPA/2020/000168]. The funders had no role in the study design, data collection, interpretation, or decision to submit the manuscript (communication No. 021/2023) for publication. The authors acknowledge all colleagues who shared the plasmids used in this study. The authors would also like to acknowledge Prateek Arora (IISER Mohali FACS Facility) for technical help with flow cytometry, Maria Ericson (Harvard Medical School) for EM imaging, and all laboratory members of the M.S. and A.T. for helpful discussions. M.S. and A.T. would like to acknowledge their toddler, Taara, for allowing them to work after office hours.

## Author contributions

K.W., M.S. and A.T. conceived and designed the study and wrote the manuscript. K.W. performed the majority of the experiments, analyzed the results, and prepared the figures. A.S., S.P. and P.C. assisted in the standardization of the protein purification protocols, live-cell imaging experiments, and critical molecular biology reagents. G.K. assisted in the confocal image analysis, and R.R. assisted in the SARS-CoV-2 infection experiments.

## Competing interests

The authors declare no competing interests.
