## [Peer Review File · Nature Communications]

SARS-CoV-2 virulence factor ORF3a blocks lysosome function by modulating TBC1D5-dependent Rab7 GTPase cycleReviewers' Comments:

Reviewer #1:

Remarks to the Author:

In this manuscript by Kshitiz Walia et al., the authors analyzed the function and mechanism of SARS-CoV-2 encoded protein ORF3a in regulating Rab7. Among the proteins encoded by SARS-CoV-2, ORF3a efficiently inhibits lysosome and autophagy process. The host cell protein, Vps39, has been found to be targeted by ORF3a binding, which causes disruption of Vps39 related functions including HOPS assembly and Rab7-HOPS interaction. Here the authors presented another mechanism of ORF3a-Vps39 in prohibiting lysosome. ORF3a binding with Vps39 separates Rab7 from GAP TBC1D5, and also disrupts Vps39-PLEKHM1 interaction, which leads to blockage of lysosome-endosome fusion and eventually causes dysfunction of lysosomal degradation. The authors nicely used high-resolution and live imaging to analyze the vesicle, endosome and lysosome trafficking and function. The quantification is also nicely done. Basically, the manuscript nicely showed another mechanism of ORF3a inhibiting lysosome function in host cell. However, there are many findings and conclusions in this manuscript have been shown in previous literatures, for example, the endosome/lysosome localization of ORF3a, the interaction between ORF3a and Vps39, the disruption of HOPS, HOPS-Rab7 and lysosome function by ORF3a (page 5). Therefore, the novelty of this manuscript is diminished. There are several points suggested by the authors to show that ORF3a disrupts lysosome function, it is suggested that some not very related points can be deleted, like the lysosomal localization of the V1G1 subunit of the V-ATPase, the CI-M6PR recycling and the sorting of newly synthesized hydrolases. Therefore, the whole work can be focusing on Vps39. Or, the focus can be how ORF3a disrupts the pH/function of lysosome (through V1G1 subunit of the V-ATPase, the CI-M6PR recycling and the sorting of newly synthesized hydrolases).

Reviewer #2:

Remarks to the Author:

In this study the authors describe the role of the SARS-Cov-2 Orf3a protein in inhibiting lysosomal function. The mechanism underpinnings of how viral accessory proteins interfere with organelle function and homeostasis is of fundamental importance. The authors show that Orf3a in particular is recruited to endolysosomal compartments and inhibits fusion of autophagosomes/endosomes with lysosomes, delineating the mechanism that potentially contributes to lysosomal-dependent egress of viral progeny particles. While the role of Orf3a in inhibiting lysosomal function has already been reported, this study provides some additional findings to establish the molecular determinants that drive lysosomal alterations observed in SARS-CoV-2 infected cells. However, some of the findings need to be clarified better experimentally. In addition, the current scope of the study feels slightly limited.

General comments:

Previous studies have shown that Orf3 in virus infected cells can generate frame shifted variants including Orf3a along with Orf3c, Orf3d, Orf3d-2 etc. It is therefore not clear, without infections with recombinant virus lacking Orf3a, whether Orf3a dependent lysosomal alterations are physiologically relevant. The expression levels of Orf3a in cells over-expressing them might be vastly different from what happens in infection. It would therefore be useful if the authors could provide data on Orf3a with live infections to provide some evidence on what happens in infected cells. Along the same lines, the authors should also comment on whether Orf3a function has evolved for this feature of lysosomal perturbation in the different natural variants of SARS-CoV-2.

Specific comments:

1. Figs 1-3 would benefit from using complementary approaches in testing the role of Orf3a in inhibiting lysosomal function. In particular, it would be important to test mutants of Orf3a which are

not recruited to the endolysosomal compartments, especially for Rab7 activation. In addition, the Orf3a-Rab7 dependent effect needs to be tested in cells deficient in Rab7 and those reconstituted with the constitutive active/inactive GTP and GDP bound versions of it.

2. Figure 4 shows that upon infection, Rab7 levels increase, but there is no evidence that this is due to Orf3a. It is also important to show that SARS-CoV-2 infected cells express Orf3a, and compare levels of Orf3a in virus infected cells to those of the mammalian expression models that the authors have used.

3. In Figure 5L, M, given the model being proposed by the authors, it is important to also show whether cells expressing the GDP bound form of Rab7 results in lower virus egress/production. These would be performed in Rab7-deficient cells expressing either the GDP bound or the GTP bound forms of Rab7.

4. It would benefit this study to include a few different versions of Orf3a from the later circulating variants of SARS-CoV-2 with greater potential of cell to cell transmission, to test whether it correlates with Orf3a function.

Response to Reviewers Comments:

We thank the reviewers for their valuable comments and suggestions pertaining to our manuscript. Below, we provide a point-by-point response (shown in blue) to how we have addressed the concerns raised by the reviewers (shown in black).

Reviewer #1 (Remarks to the Author):

In this manuscript by Kshitz Walia et al., the authors analyzed the function and mechanism of SARS-CoV-2 encoded protein ORF3a in regulating Rab7. Among the proteins encoded by SARS-CoV-2, ORF3a efficiently inhibits lysosome and autophagy process. The host cell protein, Vps39, has been found to be targeted by ORF3a binding, which causes disruption of Vps39 related functions including HOPS assembly and Rab7-HOPS interaction. Here the authors presented another mechanism of ORF3a-Vps39 in prohibiting lysosome. ORF3a binding with Vps39 separates Rab7 from GAP TBC1D5, and also disrupts Vps39-PLEKHM1 interaction, which leads to blockage of lysosome-endosome fusion and eventually causes dysfunction of lysosomal degradation. The authors nicely used high-resolution and live imaging to analyze the vesicle, endosome and lysosome trafficking and function. The quantification is also nicely done. Basically, the manuscript nicely showed another mechanism of ORF3a inhibiting lysosome function in host cell. However, there are many findings and conclusions in this manuscript have been shown in previous literatures, for example, the endosome/lysosome localization of ORF3a, the interaction between ORF3a and Vps39, the disruption of HOPS, HOPS-Rab7 and lysosome function by ORF3a (page 5). Therefore, the novelty of this manuscript is diminished. There are several points suggested by the authors to show that ORF3a disrupts lysosome function, it is suggested that some not very related points can be deleted, like the lysosomal localization of the V1G1 subunit of the V-ATPase, the CI-M6PR recycling and the sorting of newly synthesized hydrolases. Therefore, the whole work can be focusing on Vps39. Or, the focus can be how ORF3a disrupts the pH/function of lysosome (through V1G1 subunit of the V-ATPase, the CI-M6PR recycling and the sorting of newly synthesized hydrolases).

Response: We thank the reviewer for the detailed comments and suggestions to improve the manuscript. Based on the reviewer's suggestions, we have removed the section on V-ATPase and lysosomal pH. We have also added several new experiments to address all the concerns of Reviewer #2. The new evidence presented in the manuscript now categorically establishes that ORF3a is essential for the Rab7 hyperactivation observed upon SARS-CoV-2 infection (**Fig. 4C-J**). While ORF3a binding to Vps39 is already known, we established that Vps39 binding is required for ORF3a function to mediate Rab7 hyperactivation and viral production (**Fig. 5A-G**).

We also elucidated for the first time in this study that ORF3a and Vps39 form a complex with the Rab7 GAP TBC1D5 and that this interaction disrupts TBC1D5 binding to Rab7, indicating a mechanism by which ORF3a blocks the Rab7 GTPase cycle (**Fig. 5H-L**). We found that viral production is strongly correlated with the active level of Rab7, and the GDP-bound form of Rab7 inhibits viral production (**Fig. 6**). To explore how Rab7 hyperactivation benefits virus production, we investigated the known roles of Rab7 in the retrieval of CI-M6PR from the trans-Golgi network (TGN)

and in the interaction of Rab7 with Arl8b-positive endosomes. Hyperactivation of Rab7 in ORF3a-expressing cells impaired CI-M6PR retrieval from late endosomes toward the TGN, disrupting the biosynthetic transport of newly synthesized hydrolases to lysosomes (**Fig. 7**). Furthermore, the tethering of the Rab7- and Arl8b-positive compartments was strikingly reduced upon ORF3a expression (**Fig. 8**). As SARS-CoV-2 egress requires Arl8b, these findings suggest that ORF3a-mediated hyperactivation of Rab7 serves a multitude of functions, including blocking endolysosome formation, interrupting the transport of lysosomal hydrolases, and promoting viral egress (**Fig. 9**).

Reviewer #2 (Remarks to the Author):

In this study the authors describe the role of the SARS-Cov-2 Orf3a protein in inhibiting lysosomal function. The mechanism underpinnings of how viral accessory proteins interfere with organelle function and homeostasis is of fundamental importance. The authors show that Orf3a in particular is recruited to endolysosomal compartments and inhibits fusion of autophagosomes/endosomes with lysosomes, delineating the mechanism that potentially contributes to lysosomal-dependent egress of viral progeny particles. While the role of Orf3a in inhibiting lysosomal function has already been reported, this study provides some additional findings to establish the molecular determinants that drive lysosomal alterations observed in SARS-CoV-2 infected cells. However, some of the findings need to be clarified better experimentally. In addition, the current scope of the study feels slightly limited.

General comments:

Previous studies have shown that Orf3 in virus infected cells can generate frame shifted variants including Orf3a along with Orf3c, Orf3d, Orf3d-2 etc. It is therefore not clear, without infections with recombinant virus lacking Orf3a, whether Orf3a dependent lysosomal alterations are physiologically relevant. The expression levels of Orf3a in cells over-expressing them might be vastly different from what happens in infection. It would therefore be useful if the authors could provide data on Orf3a with live infections to provide some evidence on what happens in infected cells.

Response: We appreciate the reviewer's concern and suggestions to strengthen the central conclusion of our study. To address the reviewer's comments, we confirmed whether the level of ORF3a expressed in the HeLa^{ORF3a} stable cell line was comparable to that observed upon SARS-CoV-2 infection. To this end, we procured an antibody against ORF3a and confirmed its specificity by siRNA-mediated knockdown of ORF3a in SARS-CoV-2-infected Vero E6 cells (**Supplementary Fig. S1A**). Using this antibody, we found that ORF3a levels in the HeLa^{ORF3a} stable cell line after 24 hours of doxycycline treatment (1 µg/mL) were similar to those observed in SARS-CoV-2-infected Vero E6 cell lysates (**Supplementary Fig. S1B**), confirming that the phenotypes observed in the HeLa^{ORF3a} stable cell line are physiologically relevant.

To investigate whether ORF3a-dependent lysosomal alterations are physiologically relevant, we depleted ORF3a from SARS-CoV-2-infected cells using a siRNA approach. We confirmed that the efficiency of ORF3a silencing was ~70% through western blotting (**Supplementary Fig. S1A**). Quantitative RT-PCR analysis

of the expression of the *E* and *orf1ab* SARS-CoV-2 genes (**Fig. 4E**) and intracellular levels (in cellular lysates) (bottom panels in **Fig. 4C**, and quantification are shown in **Fig. 4D**) and extracellular protein levels (in culture media) of the SARS-CoV-2 N-antigen (**Fig. 4F**) confirmed a significant reduction in SARS-CoV-2 replication in ORF3a-depleted cells. These results confirm the physiological role of ORF3a in viral pathogenesis. Furthermore, as observed by the protein levels of the autophagosomal marker protein LC3b and the autophagy substrate p62, ORF3a depletion significantly rescued the autophagic flux phenotype observed in SARS-CoV-2-infected cells (top and middle panels in **Fig. 4C**), confirming that ORF3a is required for impaired autophagic flux upon SARS-CoV-2 infection.

To test whether Rab7 hyperactivation in SARS-CoV-2-infected cells is ORF3a dependent, we analyzed the localization and activation status of Rab7 in ORF3a-depleted cells. Indeed, ORF3a knockdown during SARS-CoV-2 infection restored the size and distribution of Rab7-positive endosomes to levels similar to those observed in uninfected cells (**Fig. 4G** and **4H**). Consistent with the role of ORF3a in Rab7 activation during SARS-CoV-2 infection, we found that ORF3a depletion reduced the level of GTP-bound Rab7 in SARS-CoV-2-infected cells, which was similar to the level observed in uninfected cells (**Fig. 4I** and **4J**).

Along the same lines, the authors should also comment on whether Orf3a function has evolved for this feature of lysosomal perturbation in the different natural variants of SARS-CoV-2.

Response: We thank the reviewer for this interesting suggestion. According to the GISAID database (repository of COVID-19 sequences), the natural mutations found in SARS-CoV-2 ORF3a include Q57H/S171L (beta variant), S253P (gamma variant), S26L (delta variant), and T223I (omicron variants) (**Supplementary Fig. S3H**). To address this comment on whether the natural variants of ORF3a present in different SARS-CoV-2 variants are able to promote Rab7 activation, we characterized different mutants of ORF3a and expressed them in HeLa cells. We found that all the natural variants of ORF3a localize to LEs/lysosomes and cause the enlargement of LAMP1-positive endosomes, similar to what was observed for ORF3a (WT) (**Fig. 3H** and **Supplementary Fig. S3I**). We next tested the interaction of ORF3a with Vps39 and found that all the natural variants of ORF3a interact with Vps39 (**Supplementary Fig. S3J**). However, one of the natural variants, ORF3a (Q57H/S171L), did not co-immunoprecipitate with Vps39 as much as ORF3a (WT) (**Supplementary Fig. S3J**). Chen et al. (2021) previously demonstrated that residue S171 of ORF3a is essential for binding to Vps39, and mutation of S171 to glutamic acid (E) disrupts this binding. Finally, all of the natural variants of ORF3a increased the size and number of Rab7 puncta as well as the amount of GTP-bound Rab7 in cell lysates, similar to what was observed for ORF3a (WT) (**Fig. 3I, 3J**, and **Supplementary Fig. S3K** and **S3L**). These results suggest that the localization and function of ORF3a in lysosomes are likely conserved during the evolution of SARS-CoV-2 strains.

Specific comments:

1. Figs 1-3 would benefit from using complementary approaches in testing the role of Orf3a in inhibiting lysosomal function. In particular, it would be important to test mutants of Orf3a which are not recruited to the endolysosomal compartments,

especially for Rab7 activation. In addition, the Orf3a-Rab7 dependent effect needs to be tested in cells deficient in Rab7 and those reconstituted with the constitutive active/inactive GTP and GDP bound versions of it.

Response: We thank the reviewer for this important suggestion. The SARS-CoV-2 ORF3a contains a YXX Φ sorting motif (160–163 amino acids) and a double glycine motif (187–188 amino acids) that are required for its intracellular transport from the Golgi to endosomes and lysosomes (Cruz-Cosme et al., 2022). To investigate whether ORF3a localization to LEs/lysosomes is required for ORF3a-mediated lysosomal perturbations, we disrupted the YXX Φ sorting motif (ORF3a Y160A/V163G) and investigated lysosome morphology and function in these cells (**Supplementary Fig. S1N**). As previously reported (Cruz-Cosme et al., 2022), the mutation of the YXX Φ sorting motif blocked ORF3a export from the Golgi apparatus (**Supplementary Fig. S1O**). In contrast to that of ORF3a (WT), the expression of the ORF3a (Y160A/V163G) mutant did not cause enlargement of LAMP1-positive endosomes (**Supplementary Fig. S1P**). We corroborated these observations by quantifying the average LAMP1 puncta diameter in cells expressing ORF3a (WT) or the ORF3a (Y160A/V163G) mutant (**Supplementary Fig. S1Q**). Furthermore, we found that the ORF3a (Y160A/V163G) mutant had no significant effect on Rab7 puncta size or number or on GTP-bound Rab7 levels in cell lysates (**Supplementary Fig. S3D-G**), indicating that the presence of ORF3a on LEs/lysosomes is required for its ability to promote the activation of Rab7.

In line with this particular concern and to strengthen our central conclusion that ORF3a-mediated Rab7 hyperactivation is required for viral growth and replication, we next questioned whether the loss of ORF3a in SARS-CoV-2-infected cells is compensated by active Rab7. To this end, we expressed the constitutively GTP-locked (Q67L) form of Rab7 in ORF3a siRNA-treated SARS-CoV-2-infected cells. Indeed, we found significant rescue of SARS-CoV-2 replication when activated Rab7 was expressed in ORF3a-depleted cells (**Fig. 6K-M**). These findings suggest that ORF3a plays a key role in promoting Rab7 hyperactivation, which in turn is required for viral replication.

2. Figure 4 shows that upon infection, Rab7 levels increase, but there is no evidence that this is due to Orf3a. It is also important to show that SARS-CoV-2 infected cells express Orf3a, and compare levels of Orf3a in virus infected cells to those of the mammalian expression models that the authors have used.

Response: We thank the reviewer for this important suggestion. This question was answered in the first response. Briefly, to test whether Rab7 hyperactivation in SARS-CoV-2-infected cells is ORF3a dependent, we analyzed Rab7 localization and activation status in ORF3a-depleted cells. Indeed, ORF3a knockdown during SARS-CoV-2 infection restored the size and distribution of Rab7-positive endosomes, to levels similar to those observed in uninfected cells (**Fig. 4G and 4H**). Consistent with the role of ORF3a in Rab7 activation during SARS-CoV-2 infection, we found that ORF3a depletion reduced the level of GTP-bound Rab7 in SARS-CoV-2-infected cells, which was similar to the level observed in uninfected cells (**Fig. 4I and 4J**). To answer the second comment, we confirmed the specificity of the anti-ORF3a antibody and using this antibody, we found that the ORF3a level in the HeLa^{ORF3a} stable cell line after 24 hours of doxycycline treatment (1 μ g/mL) was similar to that observed in

SARS-CoV-2-infected Vero E6 cell lysates, confirming that the phenotypes observed in the HeLa^{ORF3a} stable cell line are physiologically relevant (**Supplementary Fig. S1B**).

3. In Figure 5L, M, given the model being proposed by the authors, it is important to also show whether cells expressing the GDP bound form of Rab7 results in lower virus egress/production. These would be performed in Rab7-deficient cells expressing either the GDP bound or the GTP bound forms of Rab7.

Response: We thank the reviewer for this important and interesting experiment. To address this comment, we first confirmed that Rab7 is required for viral replication, as depletion of Rab7 in SARS-CoV-2-infected cells reduced the total protein level of the N-antigen and reduced the expression of the SARS-CoV-2 genes (**Fig. 6A-C**). Next, we investigated whether the defect in viral replication upon Rab7 depletion can be rescued by the WT, constitutively GTP-bound (Q67L), or constitutively GDP-bound (T22N) forms of Rab7. Surprisingly, we found a strong positive correlation between the active level of Rab7 and the level of SARS-CoV-2 replication, as measured by the expression of SARS-CoV-2 genes and by measuring the intracellular and extracellular protein levels of the SARS-CoV-2 N-antigen (**Fig. 6D-F**). The rescue of the viral load by constitutively GTP-bound Rab7 (Q67L) was significantly more efficient than that by Rab7 (WT), while the expression of constitutively GDP-bound Rab7 (T22N) dramatically reduced the viral load above the effect of Rab7 gene knockdown (**Fig. 6D-F**). The reasoning for the latter could be due to the dominant-negative effect of Rab7 (T22N) on endogenous Rab7, which was still present due to partial gene knockdown.

To support these findings, we next employed chemical compounds that are characterized as either Rab7 activators (ML-098) (Surviladze et al., 2010) or Rab7 inhibitors (CID-1067700) (Agola et al., 2012). First, we tested whether these chemical compounds could activate or inhibit Rab7 in Vero E6 cells. To this end, we incubated the lysates of Vero E6 cells treated with DMSO (control), ML-098, and CID-1067700 using GST-RILP (mR7BD) and immunoblotted for the detection of Rab7. As shown in **Supplementary Fig. S4N**, treatment with ML-098 or CID-1067700 increased or decreased the active level of Rab7, respectively, in the cells. Next, we treated SARS-CoV-2-infected Vero E6 cells with these compounds and analyzed viral replication. In line with our results obtained using GTP-locked and GDP-locked forms of Rab7, we found a strong positive correlation between the active level of Rab7 and SARS-CoV-2 replication, as shown by the expression of SARS-CoV-2 genes and by the measurement of the intracellular and extracellular protein levels of the SARS-CoV-2 N-antigen (**Fig. 6G-J**).

4. It would benefit this study to include a few different versions of Orf3a from the later circulating variants of SARS-CoV-2 with greater potential of cell to cell transmission, to test whether it correlates with Orf3a function.

Response: We thank the reviewer for this interesting suggestion. According to the GISAID database (repository of COVID-19 sequences), the natural mutations found in SARS-CoV-2 ORF3a include Q57H/S171L (beta variant), S253P (gamma variant), S26L (delta variant), and T223I (omicron variants) (**Supplementary Fig. S3H**). To address this comment on whether the natural variants of ORF3a present in different

SARS-CoV-2 variants are able to promote Rab7 activation, we characterized different mutants of ORF3a and expressed them in HeLa cells. We found that all the natural variants of ORF3a localize to LEs/lysosomes and cause the enlargement of LAMP1-positive endosomes, similar to what was observed for ORF3a (WT) (**Fig. 3H** and **Supplementary Fig. S3I**). We next tested the interaction of ORF3a with Vps39 and found that all the natural variants of ORF3a interact with Vps39 (**Supplementary Fig. S3J**). However, one of the natural variants, ORF3a (Q57H/S171L), did not co-immunoprecipitate with Vps39 as much as ORF3a (WT) (**Supplementary Fig. S3J**). Chen et al. (2021) previously demonstrated that residue S171 of ORF3a is essential for binding to Vps39, and mutation of S171 to glutamic acid (E) disrupts this binding. Finally, all of the natural variants of ORF3a increased the size and number of Rab7 puncta as well as the amount of GTP-bound Rab7 in cell lysates, similar to what was observed for ORF3a (WT) (**Fig. 3H-J**, and **Supplementary Fig. S3K** and **S3L**). These results suggest that the localization and function of ORF3a on lysosomes are likely conserved during the evolution of SARS-CoV-2 strains.

Reviewers' Comments:

Reviewer #2:

Remarks to the Author:

The authors have adequately addressed all the concerns I previously raised with additional experiments. The role of Orf3a is a lot clearer with the additional data.

Response to Reviewer Comment:

Reviewer #2 (Remarks to the Author):

The authors have adequately addressed all the concerns I previously raised with additional experiments. The role of Orf3a is a lot clearer with the additional data.

Response: We thank the reviewer for appreciating our work.